# Psychosocial job characteristics and mental health: Do associations differ by migrant status in an Australian working population sample?

Xiaomin Liu[1,2], Steven J. Bowe[3], Lin Li[4,5], Lay San Too[6], Anthony D. LaMontagne[1,7]*

**1** Institute for Health Transformation, School of Health & Social Development, Deakin University, Geelong, Victoria, Australia, **2** Psychiatric Unit, First Affiliated Hospital of Kunming Medical University, Kunming, Yunnan, China, **3** Faculty of Health, Deakin Biostatistics Unit, Deakin University, Geelong, Victoria, Australia, **4** Nigel Gray Fellowship Group, Cancer Council Victoria, Melbourne, Victoria, Australia, **5** Melbourne Centre for Behaviour Change, School of Psychological Sciences, The University of Melbourne, Melbourne, Victoria, Australia, **6** Centre for Mental Health, Melbourne School of Population and Global Health, The University of Melbourne, Parkville, Victoria, Australia, **7** Centre for Health Equity, Melbourne School of Population and Global Health, The University of Melbourne, Melbourne, Victoria, Australia

* tony.lamontagne@deakin.edu.au

**Data Availability Statement:** Data from the HILDA Survey is available through the National Centre for Longitudinal Data Dataverse (Australian Government Department of Social Services):

## Abstract

Migrant workers may experience higher burdens of occupational injury and illness compared to native-born workers, which may be due to the differential exposure to occupational hazards, differential vulnerability to exposure-associated health impacts, or both. This study aims to assess if the relationships between psychosocial job characteristics and mental health vary by migrant status in Australia (differential vulnerability). A total of 8969 persons from wave 14 (2014–2015) of the Household Income and Labour Dynamics in Australia Survey were included in the analysis. Psychosocial job characteristics included skill discretion, decision authority and job insecurity. Mental health was assessed via a Mental Health Inventory-5 score (MHI-5), with a higher score indicating better mental health. Migrant status was defined by (i) country of birth (COB), (ii) the combination of COB and English/Non-English dominant language of COB and (iii) the combination of COB and years since arrival in Australia. Data were analysed using linear regression, adjusting for gender, age and educational attainment. Migrant status was analysed as an effect modifier of the relationships between psychosocial job characteristics and mental health. Skill discretion and decision authority were positively associated with the MHI-5 score while job insecurity was negatively associated with the MHI-5 score. We found no statistical evidence of migrant status acting as an effect modifier of the psychosocial job characteristic—MHI-5 relationships. With respect to psychosocial job characteristic—mental health relationships, these results suggest that differential exposure to job stressors is a more important mechanism than differential vulnerability for generating occupational health inequities between migrants and native-born workers in Australia.

https://www.dss.gov.au/our-responsibilities/families-and-children/programmes-services/the-household-income-and-labour-dynamics-in-australia-hilda-survey.

**Funding:** This study is funded by Deakin University Postgraduate Research Scholarship. The funders had no role in study design, data collection and analysis, decision to publish, or preparation of the manuscript.

**Competing interests:** The authors have declared that no competing interests exist.

## Introduction

Occupational health inequities (OHIs) refer to avoidable differences in occupational exposures or work-related health outcomes between different working population groups [1,2]. Disadvantaged population groups, such as workers with lower socioeconomic status, females and migrant workers, may be more likely to experience OHIs [1,3]. According to an OHI framework developed by Landsbergis *et al*. (1) (see S1 Fig), this may be due to disadvantaged population groups having higher risks of adverse occupational exposures, they may be more vulnerable to exposure-associated health impacts, or both.

Migrant workers are more likely to be engaged in low-skilled and so-called '3D' jobs (dirty, dangerous and demeaning), signifying that they often come with high exposures to occupational hazards, including physical, chemical, and biological factors, as well as adverse psychosocial job characteristics [4,5]. Furthermore, migrant workers may have a higher vulnerability to adverse occupational exposures due to many disadvantages, such as language difficulty, culture shock, lack of social and family support and lack of access to health care. These may result in migrant workers having higher risks of work-related accidents, injuries, fatalities, physical and mental health problems than native-born workers [6,7]. Hence, migrant workers, in general, are characterised as more likely to experience the excess burden of occupational injury and illness compared to native-born workers, or OHIs [1,3,8].

Mental health problems are common in working-age populations and result in a heavy health burden [9,10]. While work is generally considered to be good for mental health, poor working conditions—including poor psychosocial work environments can result in mental health problems or may worsen existing problems [9]. Psychosocial job characteristics refer to 'factors involved with psychosocial processes linked to the social environment of work that may be important in the causation of illness' [11]; these are common exposures to which all workers may potentially be exposed. Adverse psychosocial job characteristics—including high job demands, low job control, high job strain, high job insecurity, effort-reward imbalance and lack of social support at work—have been shown to be associated with a wide range of adverse mental health outcomes, such as depression, anxiety, burnout, low life satisfaction, and suicidality and suicide mortality [12–17]. However, the vulnerability to psychosocial job characteristic-associated mental health impacts may vary between working population groups. There is some precedent in observing that those who are disadvantaged are more vulnerable; for example, compared to workers with higher socioeconomic status, those with lower socioeconomic status showed stronger associations between effort-reward imbalance and depression, as well as job strain and depression [18]. In addition, stronger associations between job insecurity and physical health outcomes, such as cardiovascular disease, among migrant versus native-born workers have been reported [1].

It has been reported that migrant workers have a higher prevalence of mental health problems, such as depression, anxiety, substance abuse, sleep disorders and burnout than native-born workers [1,7,8,19–21]. Previous studies have shown that migrant workers are more likely to experience adverse psychosocial job characteristics, such as job strain [22], lower skill discretion and job complexity [23] and higher job insecurity [24] than native-born workers. However, only a small number of previous studies have examined migrant status-based differences in psychosocial job characteristic—mental health associations internationally and in Australia [8,25–27], and the few studies to date are inconsistent as to whether migrant compared to native-born workers are more vulnerable to psychosocial job characteristics in relation to mental health [8]: some studies reported that the associations between psychosocial job characteristics and psychological distress were similar between migrant and native-born

workers [28], while some other studies found that the associations were stronger among native-born workers [29,30].

There are various possible explanations for these inconsistencies, including simplistic characterisations of migrant status. Most previous studies have defined migrants only based on country of birth. However, language proficiency and years since arrival in the host country may influence migrants' working experiences. For example, migrants from Non-English-speaking countries and recently arrived migrants are more likely to experience higher job insecurity than Australian-born workers, but not migrants from Main-English-speaking countries and migrants who arrived in Australia more than 11 years previously [24]. Limited English proficiency is an important barrier to accessing good jobs among migrants in English-speaking countries [31], and—once employed—limited English proficiency is also a barrier to getting help and support, making such workers more susceptible to isolation and discrimination at work [32]. Besides language barriers, new migrants may face additional challenges such as culture shock, lack of social connections and support resources [33], lack of recognition of overseas qualifications [34], and more. These factors could moderate psychosocial job characteristic—mental health associations among migrants.

The number of migrants in Australia is increasing, and nearly 60% are employed [35]. Migrant workers in Australia have been shown to experience lower skill discretion/complexity and higher job insecurity than Australian-born workers, especially those from Non-English-speaking countries and those recently arrived [23,24]. However, whether migrants are differentially affected by psychosocial job characteristic-associated impacts on mental health compared to Australian-born workers is unclear, especially when language proficiency and years since arrival in the host country are taken into account. Accordingly, in this study, we use an Australian working population-representative sample to assess if the relationships between psychosocial job characteristics (skill discretion, decision authority and job insecurity) and mental health vary by migrant status, taking English proficiency and years since arrival in Australia into account. Based on the premise that migrant workers are a disadvantaged population group and thus may be more vulnerable to psychosocial job characteristic-associated mental health impacts, we hypothesise that the psychosocial job characteristic—mental health relationships will be stronger for migrants, especially migrants from Non-English-speaking countries, and migrants recently arrived in Australia.

## Materials and methods

### Data source and study participants

The data come from the Household Income and Labour Dynamics in Australia Survey (HILDA), a nationally representative sample of Australian households. Data collection began in 2001 and is conducted annually [36]. It covers a wide range of information on family life, household composition, income, labour market activity, employment, socioeconomic status and health. All household members aged 15 and older were interviewed through face-to-face or telephone interviews. In wave 1 (2001), 13969 persons responded to the survey, and the response rate was 92.3%. In 2011, an additional 4280 persons were added to top up the sample size to allow a better representation of the Australian population. Wave 14 (2014–2015) was used in the current study. In this wave, 17325 individuals responded to the survey, and the response rate for the parent sample was 80.8% [37]. The final analytic sample consists of 8969 respondents. More information on the analytic sample for this paper is described under 'Inclusion and exclusion criteria' below.

## The outcome variable (mental health)

Mental health was measured using the Mental Health Inventory-5 (MHI-5), a subscale of the 36-Item Short-Form Health Survey (SF-36). SF-36 is a widely used self-completion measure of health status which has shown good validity in the Australian population [38]. The MHI-5 consists of five six-point Likert items, which have been shown to be an effective screening tool for depression and anxiety disorders in both the general population and clinical settings [39,40]. MHI-5 scores were computed using the 'sf36' command in Stata 15.1 [41], with higher scores indicating better mental health. In this study, MHI-5 ranged from 0 to 100, with an average score of 75.0 ($SD$ = 16.0, Cronbach's $\alpha$ = 0.98).

## Exposure variables (psychosocial job characteristics)

Psychosocial job characteristics were the exposure variables, which were assessed based on the Job Demand-Control model [42]. Skill discretion, decision authority and job insecurity were measured in this study.

Skill discretion and decision authority are two subscales of job control [42], which were measured by five items in HILDA [43]. Skill discretion assesses the opportunity for skill use and was measured by two items: 'My job often requires me to learn new skills' and 'I use many of my skills and abilities in my current job' (Cronbach's $\alpha$ = 0.70). Decision authority assesses the opportunity for control and was measured by three items: 'I have a lot of freedom to decide how I do my own work', 'I have a lot of say about what happens on my job' and 'I have a lot of freedom to decide when I do my work' (Cronbach's $\alpha$ = 0.84). All the items were scored from 1 'strongly disagree' to 7 'strongly agree' and have been shown to have good internal consistency in previous Australian studies [23,43]. The score of skill discretion and decision authority was computed by summing the two/three items running from 2 to 14 and from 3 to 21, respectively, with a higher score representing higher skill discretion and decision authority.

Job insecurity refers to 'the perceived threat of job loss and the worries related to that threat' [44], which was assessed by three items with the same 7-point Likert scale: 'I have a secure future in my job', 'The company I work for will still be in business 5 years from now' and 'I worry about the future of my job'. The first two items were reversed so that a higher score indicated higher job insecurity. The score of job insecurity was computed by summing the three items running from 3 to 21, with a higher score representing higher job insecurity (Cronbach's $\alpha$ = 0.67). This scale has been shown to have good internal consistency in previous Australian studies [16,24,45,46].

In this study, psychosocial job characteristics were treated as continuous to maximise statistical power.

## Potential confounders

Potential confounders included gender (binary: male and female), age (five categories: 15–24, 25–34, 35–44, 45–54, 55–64) and educational attainment (four categories: high school or lower, diploma or certificate, bachelor and postgraduate) [1,43].

## Hypothesised effect modifiers

Migrant status was the potential effect modifier, which was characterised using three measures. The first was a binary measure based on country of birth (COB) only: coded as Australian-born vs. overseas-born. The second was based on COB and the dominant language of the COB: a three-category variable coded as Australian-born, born in a Main-English-speaking country (Main-ESC-born) and born in a Non-English-speaking country (Non-ESC-born).

The third measure was based on both COB and years since arrival in Australia; it included four categories: (1) Australian-born; and overseas-born workers of those who (2) arrived ≤5 years, (3) arrived 6–10 years and (4) arrived ≥11 years previously.

## Inclusion and exclusion criteria

The following describes the respondents included into our analysis: aged 15 to 64 years; employed; provided information on all potential confounders and migrant status, including COB, dominant language of COB and the year arrived in Australia; responded to all the items for the skill discretion scale, decision authority scale and job insecurity scale; and had MHI-5 score in wave 14.

The parent sample consisted of 17325 individuals in wave 14. As shown in Fig 1, 15231 of them were aged 15–64 years and of these, 10575 were employed, 10569 provided information of all confounders and migrant status. Among these 10569, 8983 observations had complete items for the three psychosocial job characteristic scales, and 8969 also had MHI-5 score and, thus, were included in the analysis.

## Missing data

There were 1606 respondents, accounting for 15.2% of the employed sample excluded from our analysis—among whom the vast majority (n = 1586) were due to missing items of psychosocial job characteristic scale. Based on the reason for missing data provided by HILDA, 1184 among the 1586 were due to non-response to the (entire) self-completion questionnaire (SCQ) of the HILDA survey, where psychosocial job characteristics were measured. Other reasons included the SCQ not being asked (n = 286), multiple responses to the SCQ (n = 21) and refused/not stated (n = 95). Among the 95 who refused to answer, 9 were missing all items and 86 were missing one or two items for the three psychosocial job characteristic scales. The number who were missing one or two items for the psychosocial job characteristic scales (n = 86) was very small compared to the number who were missing all items (n = 1184+286+9 = 1479). Thus, there were limited possibilities for value substitution (e.g., mean substitution of single missing items).

Compared to the observations included in the analysis, the exclusions were more likely to be male (56.4% vs. 50.7%), younger (15–24 years: 24.3% vs. 17.8%), lower educated (high school or lower: 38.7% vs. 33.7%), lower in skill level (low skill level: 18.4% vs. 14.0%; high skill level: 32.6% vs. 38.7%), casually employment (23.8% vs. 19.0%) and Non-ESC-born migrants (13.8% vs. 10.3%) (all p-values <0.001).

## Statistical analysis

First, we used chi-squared tests to compare socio-demographic characteristics by migrant status. Second, linear regression was conducted to test the relationships between psychosocial job characteristics and mental health, followed by including gender, age and educational attainment into the regression model one by one, and then simultaneously, to assess the potential for confounding. Third, effect measure modification of the job characteristic—mental health relationship by migrant status was tested by fitting product terms between psychosocial job characteristics and migrant status and comparing model fit between the model with versus without the product term using the Likelihood Ratio (LR) test. Considering that LR tests only provide measures of statistical significance without information on the relative magnitude and direction of differences in exposure-outcome relationships across effect modifier groups, comparative model fit testing was complemented for descriptive/explanatory purposes by graphing the relationships between psychosocial job characteristics and mental health by migrant status

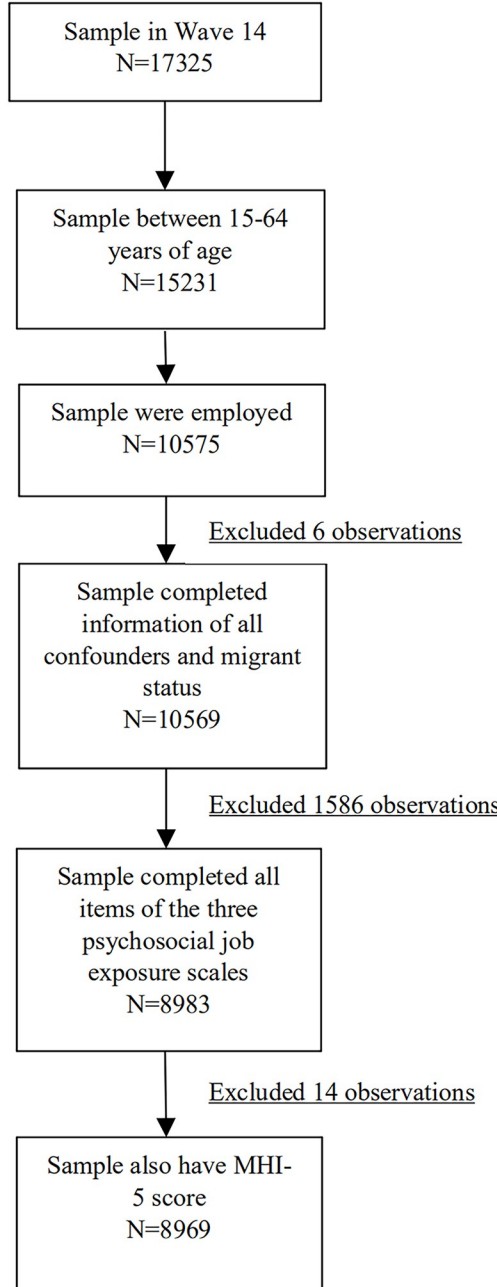

**Fig 1. Participant flow into the analytic sample.**

with 95% confidence intervals (CIs). In addition, the differences in the slope estimates of regression lines between effect modifier groups were assessed by average marginal effects comparisons [45]. LR test results or differences in slope estimates at p<0.05 was considered to constitute statistical evidence of effect measure modification and would justify the stratified presentation of job characteristic–mental health relationships as main findings. All analyses were conducted using Stata 15.1 [41].

## Sensitivity analysis

Some employment-related factors—including contract type, full/part-time and occupational skill level—were potential confounders of the relationship between psychosocial job characteristics and mental health [43,47,48]. However, contract type and occupational skill level may be potential mediators of associations between educational attainment and psychosocial job characteristics, as well as mental health (see S2 Fig); thus, controlling them in addition to educational attainment may result in an over-adjustment. Furthermore, full/part-time overlaps with working hours and might be better treated as a job characteristic. We would like to include a minimal number of confounders because interactions were tested in this study—which is often under-powered, and unnecessary covariates would further reduce the statistical power. Therefore, we conducted sensitivity analyses to assess whether our main findings differed with employment-related factors included or not (by including them one by one in multiple linear regression models, and then, simultaneously with gender, age and educational attainment).

Contract type was a four-category variable: permanent contract, casual employment, fix-term contract and self-employment. Full-time workers were those who work $\geq$35 hours per week. Occupational skill level included four categories: highest skill, mid-high skill, mid-low skill and lowest skill [49].

## Ethics approval

This study was approved by the Deakin University Human Research Ethics Committee (No. 2017–226). HILDA data access was authorised by the Australian Government Department of Social Services (DSS); therefore, consent was not obtained.

## Results

### The characteristics of respondents

Table 1 shows the characteristics of the analytic sample. A total of 8969 employed people in wave 14 were included in our analysis, among whom 1731 (19.3%) were born overseas. The differences in gender, age and educational attainment between migrant workers and Australian-born workers were apparent (all p-values <0.001). For example, Main-ESC-born workers included a significantly higher proportion of males than Australian-born workers (57.3% vs. 50.1%). Migrants who arrived in Australia between 6 to 10 years previously included a significantly higher proportion of workers with a postgraduate degree compared to Australian-born workers (30.2% vs. 11.8%). In contrast, overseas-born workers included a significantly lower proportion of workers between 15 to 24 years of age than Australian-born workers (6.3% vs. 20.5%).

Distributions of psychosocial job characteristics and MHI-5 scores by migrant status are reported in supporting information S1 Table.

### The associations between psychosocial job characteristics and mental health

Table 2 shows that, in the unadjusted model of the main analysis, skill discretion and decision authority were positively associated with the MHI-5 score, while job insecurity was negatively associated with the MHI-5 score. With every unit increase in both skill discretion and decision authority, the MHI-5 score increased 0.4, respectively; and every unit increase in job insecurity would result in a 1.0 decrease in MHI-5 score. The magnitude of association for job insecurity was much larger—nearly 2.5 times that of skill discretion and decision authority. Controlling

**Table 1. Descriptive statistics on socio-demographic characteristics by migrant status (n = 8969).**

| | Australian-born, n (%) | Migrant workers | | | | | |
|---|---|---|---|---|---|---|---|
| | | Overseas-born, n (%) | Main-ESC-born[†], n (%) | Non-ESC-born[†], n (%) | Arrived ≤5 years, n (%) | Arrived 6–10 years, n (%) | Arrived ≥11 years, n (%) |
| **Overall** | 7238 (80.70) | 1731 (19.30) | 804 (8.96) | 927 (10.33) | 154 (1.72) | 199 (2.22) | 1378 (15.36) |
| **Gender** | | | | | | | |
| Male | 3626 (50.10) | 919 (53.09) | 461 (57.34) | 458 (49.41) | 76 (49.35) | 111 (55.78) | 732 (53.12) |
| Female | 3612 (49.90) | 812 (46.91) | 343 (42.66) | 469 (50.59) | 78 (50.65) | 88 (44.22) | 646 (46.88) |
| **Age (years)** | | | | | | | |
| 15–24 | 1486 (20.53) | 109 (6.30) | 45 (5.60) | 64 (6.90) | 29 (18.83) | 25 (12.56) | 55 (3.99) |
| 25–34 | 1662 (22.96) | 370 (21.37) | 134 (16.67) | 236 (25.46) | 84 (54.55) | 81 (40.70) | 205 (14.88) |
| 35–44 | 1498 (20.70) | 416 (24.03) | 183 (22.76) | 233 (25.13) | 29 (18.83) | 64 (32.16) | 323 (23.44) |
| 45–54 | 1582 (21.86) | 493 (28.48) | 269 (33.46) | 224 (24.16) | 8 (5.19) | 25 (12.56) | 460 (33.38) |
| 55–64 | 1010 (13.95) | 343 (19.82) | 173 (21.52) | 170 (18.34) | 4 (2.60) | 4 (2.01) | 335 (24.31) |
| **Educational attainment** | | | | | | | |
| High school or lower | 2621 (36.21) | 398 (22.99) | 218 (27.11) | 180 (19.42) | 28 (18.18) | 38 (19.10) | 332 (24.09) |
| Diploma or certificate | 2530 (34.95) | 553 (31.95) | 302 (37.56) | 251 (27.08) | 38 (24.68) | 44 (22.11) | 471 (34.18) |
| Bachelor | 1232 (17.02) | 405 (23.40) | 142 (17.66) | 263 (28.37) | 50 (32.47) | 57 (28.64) | 298 (21.63) |
| Postgraduate | 855 (11.81) | 375 (21.66) | 142 (17.66) | 233 (25.13) | 38 (24.68) | 60 (30.15) | 277 (20.10) |

[†] Main-ESC-born: Born in a Main-English-speaking country; Non-ESC-born: Born in a Non-English-speaking country.

for gender, age and educational attainment either separately (see S2 Table) or simultaneously only caused small changes in the coefficients.

## The modifying role of migrant status in the relationships between psychosocial job characteristics and mental health

Likelihood ratio test results were uniformly null, providing no statistical evidence for any of our three measures of migrant status as effect modifiers of the job characteristic−mental health relationships (Skill discretion × COB: p = 0.51; skill discretion × dominant language of COB: p = 0.67; skill discretion × years since arrival: p = 0.72; decision authority × COB: p = 0.90; decision authority × dominant language of COB: p = 0.46; decision authority × years since arrival: p = 0.55; job insecurity × COB: p = 0.64; job insecurity × dominant language of COB: p = 0.80; job insecurity × years since arrival: p = 0.70).

**Table 2. Psychosocial job characteristics and mental health: Unadjusted and adjusted linear regression results (n = 8969).**

| Psychosocial job characteristics | Unadjusted | | Fully adjusted[@] | |
|---|---|---|---|---|
| | Constant value | Coefficient (95%CI) | Constant value | Coefficient (95%CI) |
| **Skill discretion** | 71.01 | 0.40 (0.28, 0.52) *** | 71.02 | 0.40 (0.28, 0.53) *** |
| **Decision authority** | 69.91 | 0.41 (0.33, 0.48) *** | 70.71 | 0.36 (0.29, 0.44) *** |
| **Job insecurity** | 83.92 | -1.04 (-1.13, -0.96) *** | 83.97 | -1.08 (-1.16, -0.99) *** |

Results of univariable and multivariable linear regressions; Mental health was measured by MHI-5 score;

*** $p < 0.001$;

[@] Fully adjusted model adjusted for gender, age and educational attainment simultaneously.

**Table 3. Relationships between skill discretion and mental health stratified by three measures of migrant status: The slope estimate within each stratum of migrant status and the differences in the slope estimates between groups, controlled for gender, age and educational attainment.**

| | Predicted mental health (MHI-5 score) | | | | | |
| --- | --- | --- | --- | --- | --- | --- |
| | Migrant status measure one | | Migrant status measure two | | Migrant status measure three | |
| **Slope estimates within the stratum of migrant status** | **Migrant status** | **Coef. (95% CI)** | **Migrant status** | **Coef. (95% CI)** | **Migrant status** | **Coef. (95% CI)** |
| | Australian-born | 0.38 (0.24, 0.51)*** | Australian-born | 0.38 (0.24, 0.51)*** | Australian-born | 0.38 (0.24, 0.51)*** |
| | Overseas-born | 0.48 (0.20, 0.75)*** | Main-ESC-born[†] | 0.37 (-0.04, 0.78) | Arrived ≤5 years | 0.10 (-0.76, 0.95) |
| | | | Non-ESC-born[†] | 0.55 (0.19, 0.92)** | Arrived 6–10 years | 0.59 (-0.16, 1.34) |
| | | | | | Arrived ≥11 years | 0.51 (0.20, 0.83)*** |
| **Differences in slope estimates** | Overseas-born vs. Australian-born | 0.10 (-0.20, 0.41) | Main-ESC-born[†] vs. Australian-born | -0.01 (-0.44, 0.43) | Arrived ≤5 years vs. Australian-born | -0.28 (-1.15, 0.58) |
| | | | Non-ESC-born[†] vs. Australian-born | 0.18 (-0.21, 0.57) | Arrived 6–10 years vs. Australian-born | 0.21 (-0.55, 0.98) |
| | | | Non-ESC-born[†] vs. Main-ESC-born[†] | 0.18 (-0.37, 0.74) | Arrived ≥11 years vs. Australian-born | 0.13 (-0.21, 0.48) |
| | | | | | Arrived 6–10 years vs. Arrived ≤5 years | 0.49 (-0.64, 1.63) |
| | | | | | Arrived ≥11 years vs. Arrived ≤5 years | 0.42 (-0.49, 1.32) |
| | | | | | Arrived ≥11 years vs. Arrived 6–10 years | -0.08 (-0.89, 0.74) |

Migrant status measure one: Country of birth (COB), measure two: COB and English/Non-English dominant language of COB, measure three: COB and years since arrival;

[†] Main-ESC-born: Born in a Main-English-speaking country; Non-ESC-born: Born in a Non-English-speaking country; Coef. was the beta coefficient;

*** p<0.001,

**p<0.01,

*p<0.05.

When graphing effect measure modification results, it appeared that slopes for migrants who had been resident for ≤5 years might differ from other groups for both decision authority and skill discretion analyses (see S3 and S4 Figs; slope estimates are reported in Tables 3 and 4). We would also note the recently arrived migrant group (≤5 years resident) was the smallest of all migrant subgroups analysed (n = 154, Table 1), and thus yielded the least precise beta/ slope estimates. Consistent with the LR test results, Tables 3 and 4 show that all of the 95% CIs comparing slope differences for skill discretion and decision authority analyses included zero.

In the job insecurity—mental health relationships, the regression lines for all groups, including Australian-born workers and all migrant subgroups based on COB, dominant language of COB and years since arrival in Australia, were almost overlapping (see S5 Fig; slope estimates are reported in Table 5). Consistently, Table 5 shows none of the differences in slope estimates between groups was significantly different from zero.

In summary, there was no statistical evidence supporting any of our three measures of migrant status as effect modifiers of the relationships between psychosocial job characteristics and mental health, despite the suggestion of some differences by visual inspection of graphical results. The main finding of our effect measure modification analysis is that the relationship between the three job characteristics and mental health does not differ by any of the three measures of migrant status used.

**Table 4. Relationships between decision authority and mental health stratified by three measures of migrant status: The slope estimate within each stratum of migrant status and the differences in the slope estimates between groups, controlled for gender, age and educational attainment.**

| | Predicted mental health (MHI-5 score) | | | | | |
|---|---|---|---|---|---|---|
| | Migrant status measure one | | Migrant status measure two | | Migrant status measure three | |
| **Slope estimates within the stratum of migrant status** | **Migrant status** | **Coef. (95% CI)** | **Migrant status** | **Coef. (95% CI)** | **Migrant status** | **Coef. (95% CI)** |
| | Australian-born | 0.36 (0.28; 0.44)*** | Australian-born | 0.36 (0.28; 0.44)*** | Australian-born | 0.36 (0.28; 0.44)*** |
| | Overseas-born | 0.37 (0.21; 0.54)*** | Main-ESC-born[†] | 0.27 (0.03; 0.50)* | Arrived ≤5 years | -0.04 (-0.73; 0.65) |
| | | | Non-ESC-born[†] | 0.47 (0.24; 0.70)*** | Arrived 6–10 years | 0.58 (0.08; 1.08)* |
| | | | | | Arrived ≥11 years | 0.37 (0.19; 0.55)*** |
| **Differences in slope estimates** | Overseas-born vs. Australian-born | 0.01 (-0.17; 0.19) | Main-ESC-born[†] vs. Australian-born | -0.09 (-0.34; 0.15) | Arrived ≤5 years vs. Australian-born | -0.40 (-1.10; 0.29) |
| | | | Non-ESC-born[†] vs. Australian-born | 0.11 (-0.13; 0.36) | Arrived 6–10 years vs. Australian-born | 0.22 (-0.28; 0.73) |
| | | | Non-ESC-born[†] vs. Main-ESC-born[†] | 0.21 (-0.12; 0.53) | Arrived ≥11 years vs. Australian-born | 0.01 (-0.18; 0.21) |
| | | | | | Arrived 6–10 years vs. Arrived ≤5 years | 0.62 (-0.23; 1.48) |
| | | | | | Arrived ≥11 years vs. Arrived ≤5 years | 0.42 (-0.30; 1.13) |
| | | | | | Arrived ≥11 years vs. Arrived 6–10 years | -0.21 (-0.74; 0.32) |

Migrant status measure one: Country of birth (COB), measure two: COB and English/Non-English dominant language of COB, measure three: COB and years since arrival;

[†] Main-ESC-born: Born in a Main-English-speaking country; Non-ESC-born: Born in a Non-English-speaking country; Coef. was the beta coefficient;

*** p<0.001,

**p<0.01,

*p<0.05.

## Sensitivity analysis

Based on the results of sensitivity analysis (see S2 Table), including contract type, full/part-time and occupational skill level, either separately or simultaneously, into the linear regressions only resulted in small changes in the coefficients; therefore, they were excluded from the main analysis in this study.

## Discussion

Inconsistent with our hypothesis, the magnitude of associations between psychosocial job characteristics and mental health did not vary by migrant status in our working Australian population sample despite investigating migrant status with three distinct measures. However, there was a suggestion that psychosocial job characteristic−mental health associations for migrants who had arrived in Australia ≤5 years previously differed from the associations for Australian-born workers and migrants who had been living in Australia for longer, though we could not rule out the possibility that this difference arose by chance. Skill discretion and decision authority were positively associated with MHI-5 score for both Australian-born workers and most subgroups of migrants, but there was suggestive evidence that skill discretion and decision authority may not be associated with MHI-5 score for the most recently arrived

**Table 5. Relationships between job insecurity and mental health stratified by three measures of migrant status: The slope estimate within each stratum of migrant status and the differences in the slope estimates between groups, controlled for gender, age and educational attainment.**

| | Predicted mental health (MHI-5 score) | | | | | |
| | Migrant status measure one | | Migrant status measure two | | Migrant status measure three | |
| **Slope estimates within the stratum of migrant status** | **Migrant status** | **Coef. (95% CI)** | **Migrant status** | **Coef. (95% CI)** | **Migrant status** | **Coef. (95% CI)** |
|---|---|---|---|---|---|---|
| | Australian-born | -1.08 (-1.18; -1.00)*** | Australian-born | -1.08 (-1.18; -1.00)*** | Australian-born | -1.08 (-1.18; -1.00)*** |
| | Overseas-born | -1.03 (-1.21; -0.85)*** | Main-ESC-born[†] | -0.99 (-1.26; -0.71)*** | Arrived ≤5 years | -0.79 (-1.44; -0.14)* |
| | | | Non-ESC-born[†] | -1.07 (-1.31; -0.83)*** | Arrived 6–10 years | -1.27 (-1.80; -0.74)*** |
| | | | | | Arrived ≥11 years | -1.03 (-1.24; -0.83)*** |
| **Differences in slope estimates** | Overseas-born vs. Australian-born | 0.05 (-0.15; 0.25) | Main-ESC-born[†] vs. Australian-born | 0.10 (-0.19; 0.39) | Arrived ≤5 years vs. Australian-born | 0.29 (-0.37; 0.95) |
| | | | Non-ESC-born[†] vs. Australian-born | 0.01 (-0.25; 0.27) | Arrived 6–10 years vs. Australian-born | -0.18 (-0.72; 0.35) |
| | | | Non-ESC-born[†] vs. Main-ESC-born[†] | -0.09 (-0.45; 0.28) | Arrived ≥11 years vs. Australian-born | 0.05 (-0.17; 0.27) |
| | | | | | Arrived 6–10 years vs. Arrived ≤5 years | -0.48 (-1.32; 0.36) |
| | | | | | Arrived ≥11 years vs. Arrived ≤5 years | -0.24 (-0.93; 0.44) |
| | | | | | Arrived ≥11 years vs. Arrived 6–10 years | 0.23 (-0.33; 0.80) |

Migrant status measure one: Country of birth (COB), measure two: COB and English/Non-English dominant language of COB, measure three: COB and years since arrival;

[†] Main-ESC-born: Born in a Main-English-speaking country; Non-ESC-born: Born in a Non-English-speaking country; Coef. was the beta coefficient;

*** $p < 0.001$,

**$p < 0.01$,

*$p < 0.05$.

migrants. Job insecurity was negatively associated with the MHI-5 score for both Australian-born workers and all subgroups of migrant workers. Furthermore, job insecurity was a stronger determinant of mental health than skill discretion and decision authority in all workers, with a larger magnitude of effect on MHI-5 score.

Most previous studies, internationally, found either that migrants and native-born workers have similar psychosocial job characteristic−mental health associations or native-born workers have even stronger associations [8,26,28]. For example, Aalto et al. (28) reported that psychosocial job characteristics were not associated with mental health for both migrant and native-born workers in Finland. Moreover, Font et al. (29) and Ortega et al. (30) found that adverse psychosocial job characteristics led to poor mental health for both migrants and native-born workers, and the magnitudes of associations were larger for native-born workers. However, these two studies compared the psychosocial job characteristic−mental health associations separately among migrants and native-born workers without testing whether the differences in the associations between migrants and native-born workers were statistically significant or not. In our study, even when we take dominant language of COB and years since arrival in Australia into account, we still do not find significant statistical evidence that relationships between psychosocial job characteristics and mental health are different between migrant and native-born workers. We did, however, find suggestive evidence that skill

discretion and decision authority may not be associated with the mental health of most recently arrived migrants, suggesting associations for these two psychosocial job characteristics are stronger in Australian-born workers (see Tables 3 and 4 and S3 and S4 Figs); however, these apparent differences may be due to the small sample size of the subgroup of migrants who arrived in Australia ≤5 years previously (n = 154). Future study with a larger sample size would be required to resolve this question.

Various explanations as to why migrants would not have a higher vulnerability to psychosocial job characteristics than native-born workers are plausible. Some factors, such as 'healthy immigrant effect' [29], migrants' expectations, cultural characteristics of migrants' original countries [50] and previous working conditions may buffer the migrants' vulnerability to adverse psychosocial job characteristics. Combining these factors with years since arrival in host countries, there could be acculturation-related changes in vulnerability to adverse psychosocial job characteristics. These factors may act as buffers for the first few years of post-arrival. For example, apart from 'healthy immigrant effect', migrants may have the psychological preparation that they may have little chance to use their skills or that they will need to change jobs frequently, so their expectations are low. In addition, as Hoppe (50) indicated, migrants from some countries where they always have lower job control may not feel the need for job control; or comparing with the working conditions in their original countries, the working conditions in the new country may be much better. As residence time increases, such as after ten years post-arrival, when migrants are reported as achieving similar labour market outcomes as native-born workers [51], many of the difficulties that new migrants encountered may be overcome, but the 'healthy immigrant effect' may disappear and migrants' occupational expectations may increase as well. Therefore, determinants of migrants' vulnerability may be close to those of native-born workers due to acculturation as years since arrival increases.

In our previous studies of the same sample [23], we found that migrants from Non-English-speaking countries experience lower skill discretion than Australian-born workers even more than 11 years post-arrival in Australia, while migrants and Australian-born workers have similar decision authority. Furthermore, migrants from Non-English-speaking countries have higher job insecurity than Australian-born workers up to 11 years post-arrival [24]. In particular, highly educated migrants from Non-English-speaking countries with bachelor or postgraduate educational level experience 'double adverse exposures' (lower skill discretion and higher job insecurity) even after 11 years post-arrival in Australia [23,24]. These findings suggested that despite the lack of statistical evidence of effect measure modification of migrant status on psychosocial job characteristic—mental health associations, disparities in exposure still prevail. Poorer psychosocial job characteristics for migrant workers, especially those from Non-English-speaking countries and arrived in Australia ≤11 years previously would still contribute to OHIs between migrant and Australian-born workers.

Our findings may suggest some intervention strategies to reduce OHIs among Australian migrant workers. Based on the results indicating that exposure to job insecurity may have the strongest association with poor mental health outcomes [52,53], reducing job insecurity would seem to be the most appropriate target for reducing the risk of OHIs between migrant workers and Australian-born workers. Furthermore, language difficulty may contribute to the adverse psychosocial job characteristic exposures, especially for the highly educated migrants from Non-English-speaking countries [23,24]; therefore, continuing language support may be helpful to reduce OHIs experienced by migrants compared to Australian-born workers.

Some limitations should be considered in the interpretation of our findings. First, we used a cross-sectional design, precluding causal inference in the exposure-outcome relationship. However, other evidence suggests that psychosocial job characteristics are causally-dominant in relation to mental health [52–55]. Second, our results may have been biased toward the null

due to the healthy worker effect—some workers may have dropped out of the workforce due to poor working conditions thus underestimating exposure-outcome associations. Third, migrants from Non-English-speaking countries with lower English proficiency were more likely to be non-respondents on the SCQ [56]; to the extent that workers with lower English proficiency are employed in lower quality jobs, we would expect this to underestimate the exposure and possibly exposure-outcome associations. Further, non-respondents on the SCQ were more likely to be younger, less educated and lower skilled, and other evidence suggests these groups may be more vulnerable to adverse psychosocial job characteristic exposures [1,18], which again would potentially result in underestimating exposure and possibly exposure-outcome associations. Fourth, our study is limited by its reliance on self-report measures, both for psychosocial job characteristics and mental health; thus, common method bias could potentially inflate the relationship between psychosocial job characteristics and mental health. Fifth, the (ecological) variable of the dominant language of the country of birth may have included some migrant workers whose dominant language was not English, but still spoke English proficiently. This exposure misclassification would make it harder to observe the differences in psychosocial job characteristic—mental health associations between migrants from Non-English-speaking countries and both native-born workers and migrants from Main-English-speaking countries. Finally, our statistical power to assess effect measure modification was limited by the relatively small sample sizes of subgroups, particularly those migrants who reported being resident in Australia for ≤5 years.

The above limitations are offset by the particular strengths of this study. First, it presents a thorough comparison of differences in psychosocial job characteristic—mental health associations between migrant and native-born workers using a national population-representative sample. Second, it unpacks the notion of 'migrant status' and defines and analyses it in three distinct ways. Third, based on our previous studies on differential exposure to psychosocial job characteristics between migrant and native-born workers in Australia, our results identify possible intervention targets to reduce occupational mental health inequities affecting migrant workers.

## Conclusions

In conclusion, we found little evidence that the magnitudes of psychosocial job characteristic—mental health relationships differ between migrant workers and Australian-born workers. However, there was suggestive evidence that skill discretion and decision authority may not be associated with migrants' mental health during their first five years post-arrival in Australia. Because migrants from Non-English-speaking countries experience significantly lower skill discretion and higher job insecurity than Australian-born workers, reducing these adverse psychosocial job characteristic exposures—especially job insecurity and continuing English improvement practices could reduce occupational mental health inequities for migrant workers in Australia.

## Supporting information

**S1 Fig. Conceptual overview of the role of work organisation in the creation of OHIs (adapted and modified based on Landsbergis et al. 2014).**
(PDF)

**S2 Fig. Directed acyclic graph (DAG) of the relationship between psychosocial job characteristics and mental health.** PJC: Psychosocial job characteristics; MH: Mental health; EDU:

Educational attainment; Con-T: Contract type; OSL: Occupational skill level.
(PDF)

**S3 Fig. Relationships between skill discretion and mental health stratified by three measures of migrant status.** Graphs generated from linear regressions with the product terms of skill discretion and migrant status and controlled for gender, age and educational attainment. Mental health was measured by the MHI-5 score. Shown with 95% CIs.
(PDF)

**S4 Fig. Relationships between decision authority and mental health stratified by three measures of migrant status.** Graphs generated from linear regressions with the product terms of decision authority and migrant status and controlled for gender, age and educational attainment. Mental health was measured by the MHI-5 score. Shown with 95% CIs.
(PDF)

**S5 Fig. Relationships between job insecurity and mental health stratified by three measures of migrant status.** Graphs generated from linear regressions with the product terms of job insecurity and migrant status and controlled for gender, age and educational attainment. Mental health was measured by the MHI-5 score. Shown with 95% CIs.
(PDF)

**S1 Table. Distribution of psychosocial job characteristics and MHI-5 score by migrant status.** Main-ESC-born: Born in a Main-English-speaking country, Non-ESC-born: Born in a Non-English-speaking country.
(PDF)

**S2 Table. Psychosocial job characteristics and mental health: Unadjusted and adjusted linear regression results of sensitivity analysis (n = 8917).** Results of univariable and multivariable linear regressions; *** $p < 0.001$; @ Adjusted model adjusted for gender, age, educational attainment, contract type, full/part-time and occupational skill level simultaneously. Note: Controlling for contract type, full/part-time and occupational skill level in the analyses reduced the sample size to 8917 because there were 52 observations did not provide information on these three confounders.
(PDF)

## Acknowledgments

This paper uses data from the Household, Income and Labour Dynamics in Australia (HILDA) Survey. The HILDA Project is funded by the Australian Government Department of Social Services (DSS) and is managed by the Melbourne Institute of Applied Economic and Social Research at the University of Melbourne. The findings and views reported in this paper, however, are those of the authors and should not be attributed to either DSS or the Melbourne Institute. Its contents, including any opinions, and/or conclusions expressed, are solely those of the authors.

## Author Contributions

**Conceptualization:** Xiaomin Liu, Steven J. Bowe, Lin Li, Lay San Too, Anthony D. LaMontagne.

**Data curation:** Xiaomin Liu.

**Formal analysis:** Xiaomin Liu, Steven J. Bowe.

**Funding acquisition:** Xiaomin Liu, Anthony D. LaMontagne.

**Investigation:** Xiaomin Liu.

**Methodology:** Xiaomin Liu, Steven J. Bowe, Lin Li, Lay San Too, Anthony D. LaMontagne.

**Project administration:** Anthony D. LaMontagne.

**Supervision:** Steven J. Bowe, Lin Li, Lay San Too, Anthony D. LaMontagne.

**Validation:** Anthony D. LaMontagne.

**Writing – original draft:** Xiaomin Liu.

**Writing – review & editing:** Xiaomin Liu, Steven J. Bowe, Lin Li, Lay San Too, Anthony D. LaMontagne.

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
