## [Decision Letter · Decision Letter 0]

7 Apr 2020

PONE-D-19-35389

Do relationships between job stressor exposures and mental health vary by migrant status? An Australian comparative analysis

PLOS ONE

Dear Dr. LIU,

I do hope this message finds you well! 

Thank you for submitting your manuscript to PLOS ONE. We received two reviews by experts in the field, and as is the normal process at PLOS ONE, I read the manuscript independently from the two reviews. Your topic is timely, the question relevant, and the manuscript was generally well-written. However, despise these strengths, the two reviewers raised serious concerns about the paper. As the manuscript has a potential merit, we invite you to submit a revised version of the manuscript that addresses the points raised during the review process.

The Reviewers' specific comments are included below. They are detailed and constructive and I will not repeat. Rather, I have highlighted what I see as their major concerns, as well as added some comments of my own. 

It should be addressed why it is important to study migrant status as a boundary condition of the relationship between job stressors and strains. This should go along with a clear definition of job stressors and strains. Furthermore, it is important to distinguish between job stressors (e.g. job insecurity), job resources (e.g. skill discretion, decision authority, social support), and strains (e.g. deteriorated mental health). A low level of a job resource does not necessarily imply that a stressor is present. The manuscript needs to be repositioned as skill discretion and decision authority do not represent job stressors. I see two different ways to deal with this issue: On the one hand, it is possible to look at migrant status as the modifier of the relationships between job stressors and strain as well job resources and strain. On the other hand, HILDA might include other variables that are commonly understood as job stressors (e.g. workload, social conflict). The study should be presented within a theoretical framework implying that other socio-demographic differences could be either considered in the data analysis to be ruled out as potential confounders or discussed (see Reviewer 1, major concern)The conduction (see Reviewer 2, major concern) and presentation (see Reviewer 1, minor concern) of the data analysis should be revisedMore attention to detail throughout the manuscript would be warranted (see several concerns raised by Reviewer 2)

We would appreciate receiving your revised manuscript by May 22 2020 11:59PM. To enhance the reproducibility of your results, we recommend that if applicable you deposit your laboratory protocols in protocols.io, where a protocol can be assigned its own identifier (DOI) such that it can be cited independently in the future. For instructions see: http://journals.plos.org/plosone/s/submission-guidelines#loc-laboratory-protocols

We look forward to receiving your revised manuscript.

Kind regards,

Dana Unger

Academic Editor

PLOS ONE

2. In ethics statement in the manuscript and in the online submission form, please provide additional information about the patient records used in your retrospective study. Specifically, please ensure that you have discussed whether all data were fully anonymized before you accessed them and/or whether the IRB or ethics committee waived the requirement for informed consent. If patients provided informed written consent to have data from their medical records used in research, please include this information.

Reviewers' comments:

Reviewer's Responses to Questions

**Comments to the Author**

1. Is the manuscript technically sound, and do the data support the conclusions?

Reviewer #1: Partly

Reviewer #2: Partly

2. Has the statistical analysis been performed appropriately and rigorously? 

Reviewer #1: I Don't Know

Reviewer #2: I Don't Know

3. Have the authors made all data underlying the findings in their manuscript fully available?

Reviewer #1: Yes

Reviewer #2: No

4. Is the manuscript presented in an intelligible fashion and written in standard English?

Reviewer #1: Yes

Reviewer #2: Yes

5. Review Comments to the Author

Reviewer #1: Do relationships between job stressor exposures and mental health vary by migrant status? An Australian comparative analysis

Comments to the Authors

This is an interesting article on a relevant issue for occupational health / public health that focus on a topic of growing interest.

I want to congratulate the Authors for the ideas, effort and quality of their work. After reading the article in deep, I have some major and minor comments. Hopefully they will be useful to the Authors.

Major comments

For me, there is a surprising thing in this article: the absence of some key concepts (and variables) theoretically related in the literature to psychosocial exposures, mental health and migrant status: 1) “social class” (or any proxi as “occupational grade” or “socioeconomic status”..., i.e. “manual / non manual workers”, “blue / white collar”, ESEC categories..., depending on the available data and the theoretical background); and 2) employment conditions, as type of contract, i.e. fixed or temporary, partial time or full time work, level of precariousness... All of them have been related to psychosocial exposures, mental health and migrant status in other studies; but only low occupational position is mentioned once in the introduction with no consequences in the analysis and discussion.

The Authors find almost no differences on psychosocial exposures regarding migrant status; perhaps because what would explain main differences could be social class / occupational grade or employment conditions?

Even considering that the unemployed at the time of interview were excluded in this study, the study population may still be a mixture of persons employed under very different conditions that are related to exposures and outcomes, mainly if the work is temporary or permanent, and full-time or part-time; low or high grade.

The Authors mention that Australian-born workers have a much lower proportion (12 vs 25%) of workers with a postgraduate degree than migrants Non-ESC-born, but no differences in occupational class / category /grade; temporary / permanent, full / partial?

I wonder if HILDA does not include any variable that could be used at least as a proxy of these constructs to take them into account in the analysis as a potential confounders, since they may relate to both exposures and outcome; but I think these are two important issues that should be addressed in the paper, and at least mentioned in the introduction and considered in the discussion –if no possible in the analysis.

Minor comments

In my opinion, data source section of the article is quite difficult to read, with many population numbers given that may confuse the reader on the final sample size.

What is the final response rate?

I would suggest edit Table 1, making it clear that columns 3 to 7 refer only to overseas-born workers.

Reviewer #2: This manuscript assesses whether migrant workers are more vulnerable to job stressor exposures in terms of mental health, based on a survey in 2014-2015 among 8970 persons in an Australian working population-representative sample. Few previous studies have examined migrant status-based differences in job stressor—mental health associations. Therefore, whether migrants are differentially affected by job stressor exposures is unclear, especially when language proficiency and years since arrival in host country are taken into account. Thus, this topic is relevant and interesting.

The manuscript is mostly well-written. However I have some comments. The authors conclude that they found little evidence that the relationship between job stressor exposures and mental health is modified by migrant status. However, the stratified analyses, presented in the figures, show that migrants who arrived ≤5 years previously differed from other groups. Also migrants born in a Non-English speaking country seem to differ somewhat from other groups. The stratified analyses, thus, indicate that there is effect modification. This is my major comment.

Other comments are mostly spelled out in detail for the respective sections of the manuscript.

TITLE:

The title of the manuscript ("vary by migrant status") does not quite correspond with the aim of the study ("more vulnerable")

ABSTRACT:

The year of the survey would be much more informative to the reader than the wave number.

In order to understand the results, the reader would need to know that higher MHI-5 scores indicate better mental health (contrary to Hopkins symptom check List, HSCL), and I suggest adding this information to the abstract.

INTRODUCTION:

Lines 98-101: The reference to this sentence (Migrant workers…) seems to be missing.

MATERIAL AND METHODS:

It is stated that the study participants were interviewed through face-to-face or telephone interviews. However, there also seems to have been a Self-Completion Questionnaire, mentioned further down.

P. 6: Job control was measured by two subscales (line 140), however, further down one of the subscales (decision authority) is referred to as (job) control (line 143).

These were measured from 1 to 7, and seem to be a Likert scale, but this is not stated, as for the outcome and job insecurity.

P. 7: I would suggest referring to confounders and effect modifiers as POTENTIAL confounders and effect modifiers in the Methods section, before doing the analyses.

P. 8: Inclusion, exclusion and missing data: I would suggest including a flow chart showing this. In the Missing data part, it is not so clear whether the different categories of missing data are overlapping or if the numbers with missing data for each step only include those without missing data in the previous step.

Lines 191-195: The different reasons for refusing to answer (or not stated?) do not seem so interesting to me (but I may be wrong).

Lines 195-197: Were the 86 respondents excluded?

P. 9, lines 200-202: As all p-values are <0.001, I suggest writing this only once, at the end of the sentence.

The same comment applies to the presentation of the characteristics of subpopulations (P. 10, lines 228-233).

Statistical analysis:

Linear regression was used to assess the relationships between job stressors and mental health. Did you check that these relationships actually are linear, by plotting the exposure and outcome data? If they are not, it might be more correct to treat the variables as categorical data, e.g. by combing the levels into a few categories, using the highest category (for job control) and the lowest category (for job insecurity) as reference, respectively.

LR tests were used to assess migrant status as an effect modifier of the job stressor—mental health relationships. However, this may be better evaluated in stratified analyses, as commented below (Results section).

RESULTS:

Some of the results of Table 1 are presented in the text of this section, without the decimals, which is perfectly fine and, I would also suggest showing maximum 1 decimal in the table.

A couple of places in the Results section, the authors write that the results suggest (lines 245, 275) or indicate (l. 359) something. This is an interpretation of the results and should be moved to the discussion.

The value of the constant is shown in Table 2, but not mentioned in the Results or Discussion sections, as far as I can see. Do the differences in the constant value indicate important differences between the subgroups?

I would suggest writing what kind of analyses the results of the table are based on.

Based on the LR test results, the authors state that there was no statistical evidence confirming migrant status acting as effect modifiers of the job stressor exposure—mental health relationships. This is also the conclusion of the manuscript. However, the stratified analyses, presented in the figures, show that migrants who arrived ≤5 years previously differed from other groups. Also migrants born in a Non-English speaking country seem to differ somewhat from other groups. The stratified analyses, thus, indicate that there is effect modification.

To me, all three job stressors seem to have less effect on mental health among migrants who arrived ≤5 years previously, compared to other groups. They also seem to be less vulnerable to high job insecurity. The lines of the Australian-born group generally seem to lie somewhat higher than the lines of other groups for the two job control subscales. Could this indicate an interesting result?

I am not a statistician, and will not speculate why the LR test did not indicate effect modification. However, I will just mention that results of effect modification analyses will depend on the measures used, therefore, Rothman prefer to use the term "effect MEASURE modification" (e.g. in the book Modern Epidemiology), indicating that you e.g. may find effect modification on an additive scale, but not on a relative scale, or vice versa. Is it correct to test effect modification in linear analysis by adding a multiplicative interaction term? Or should this preferably be tested in other ways?

DISCUSSION:

The first sentence (lines 294-296) is not according to the results, as far as I can see, as mentioned above.

The authors refer to a study by Hoppe indicating that migrants from some countries may experience that working conditions in the new country is much better (lines 327-328). The opposite may also be true, if the migrants' education is not approved in the new country, and they have to accept manual work with lower job control and higher job insecurity.

Page 16 lines 359-361: The authors state: "…apart from reducing occupational health inequalities of migrant workers, reducing job stressor exposures should improve mental health of Australia-born workers as well."

This could easily result in occupational health inequalities being maintained.

Lines 366-368: Maybe add that poor working conditions may lead to poor health, and therefore healthy worker survivor effect (there seemed to be a missing link).

Page 17, lines 373-375: Common method bias could also be a limitation of a study based on subjective/self-reported job stressors and mental health.

REFERENCES:

When referring to more than one authors in the text, the authors include two of the names, instead of using one name + et al. Is this according to the PLOS?

For some references in the reference list, the journal titles are spelled out, while for others, the usual abbreviated forms are used. These should also be according to the style of the journal.

LANGUAGE:

The language is generally good, but some sentences starting with "there is/were…" seem a bit incomplete (e.g. lines 101, 120, 122 and 125). A couple of places there is also an S missing in 3rd person present tense (lines 105 and 143). There are also a few other things.

Line 365: Should "casual" be "causal"?

6. PLOS authors have the option to publish the peer review history of their article (what does this mean?). If published, this will include your full peer review and any attached files.

Reviewer #1: No

Reviewer #2: No

---

## [Author Response · Author response to Decision Letter 0]

21 May 2020

Academic editor

Major concerns

AE-1. It should be addressed why it is important to study migrant status as a boundary condition of the relationship between job stressors and strains. This should go along with a clear definition of job stressors and strains. Furthermore, it is important to distinguish between job stressors (e.g. job insecurity), job resources (e.g. skill discretion, decision authority, social support), and strains (e.g. deteriorated mental health). A low level of a job resource does not necessarily imply that a stressor is present. The manuscript needs to be repositioned as skill discretion and decision authority do not represent job stressors. I see two different ways to deal with this issue: On the one hand, it is possible to look at migrant status as the modifier of the relationships between job stressors and strain as well job resources and strain. On the other hand, HILDA might include other variables that are commonly understood as job stressors (e.g. workload, social conflict). 

Response:

Thank you for the comments. We recognize that there is diversity in the theoretical models relating to psychosocial work environments and health. We have adopted the Job Demand-Control (JDC) model in this study. In the JDC model, psychosocial job characteristics are treated as independent exposures, thus we treat skill discretion, decision authority and job insecurity as exposures in our study. We agree with you that the term ‘job stressors’ may result in confusion, and have changed it to ‘psychosocial job characteristics’ in this revised manuscript. Furthermore, we understand that ‘strain’ can be used to define exposure-associated ill-mental health; however, ‘strain’ as used in the term ‘job strain’ refers to the combination of high job demands and low job control in the JDC model; thus, we prefer to not use ‘strain’ in this context to avoid confusion. Moreover, as per your suggestion, we have revised our introduction to provide a clearer definition of occupational health inequalities (OHIs) and the importance of migrant status in research on OHIs.

The corresponding revisions to the text are:

Title

Do psychosocial job characteristic-associated impacts on mental health differ by migrant status? An Australian comparative analysis

Introduction (page 3-4, line 52-78)

Occupational health inequalities (OHIs) refer to avoidable differences in occupational exposures or work-related health outcomes between different occupational population groups [1, 2]. Disadvantaged population groups, such as workers with lower socioeconomic status, females and migrant workers, may be more likely to experience OHIs [1, 3]. According to an OHI framework developed by Landsbergis et al. (1) (see supporting information S1 Fig), this may be due to disadvantaged population groups having higher risks of adverse occupational exposures, or they may be more vulnerable to exposure-associated health impacts.

Migrant workers are more likely to be engaged in low-skilled and so-called ‘3D’ jobs (dirty, dangerous and demeaning), which often come with high exposures to occupational hazards, including physical, chemical, and biological factors, as well as adverse psychosocial job characteristics [4, 5]. Furthermore, migrant workers may have a higher vulnerability to adverse occupational exposures due to many disadvantages, such as language difficulty, culture shock, lack of social and family support and lack of access to health care. These may result in migrant workers having higher risks of work-related accidents, injuries, fatalities, physical and mental health problems than native-born workers [6, 7]. Hence, migrant workers, in general, are characterised as more likely to experience OHIs [1, 3, 8].

Mental health problems are common among working-age populations and result in a heavy health burden [9, 10]. While work is generally considered to be good for mental health, poor working conditions―including poor psychosocial work environments can result in mental health problems or may worsen existing problems [9]. Psychosocial job characteristics refer to ‘factors involved with psychosocial processes linked to the social environment of work that may be important in the causation of illness’ [11]; these are common exposures to which all workers may potentially be exposed. Adverse psychosocial job characteristics―including high job demands, low job control, job strain, job insecurity, effort-reward imbalance and lack of social support at work―have been shown to be associated with a wide range of adverse mental health outcomes, such as depression, anxiety, burnout, low life satisfaction, and suicidality and suicide mortality [12-16].

AE-2. The study should be presented within a theoretical framework implying that other socio-demographic differences could be either considered in the data analysis to be ruled out as potential confounders or discussed (see Reviewer 1, major concern).

Response:

Thank you for this comment. We have addressed this comment in the response to Reviewer 1 (below).

AE-3. The conduction (see Reviewer 2, major concern) and presentation (see Reviewer 1, minor concern) of the data analysis should be revised.

Response:

Thank you for this comment. We have addressed these comments in the response to Reviewers 1 and 2, respectively.

AE-4. More attention to detail throughout the manuscript would be warranted (see several concerns raised by Reviewer 2).

Response:

Thank you for this comment. We have edited and proofread the manuscript carefully.

Reviewer 1

This is an interesting article on a relevant issue for occupational health / public health that focus on a topic of growing interest. I want to congratulate the Authors for the ideas, effort and quality of their work. After reading the article in deep, I have some major and minor comments. Hopefully they will be useful to the Authors.

Major comments

R1-1. For me, there is a surprising thing in this article: the absence of some key concepts (and variables) theoretically related in the literature to psychosocial exposures, mental health and migrant status: 1) “social class” (or any proxi as “occupational grade” or “socioeconomic status”..., i.e. “manual / non manual workers”, “blue / white collar”, ESEC categories..., depending on the available data and the theoretical background); and 2) employment conditions, as type of contract, i.e. fixed or temporary, partial time or full time work, level of precariousness... All of them have been related to psychosocial exposures, mental health and migrant status in other studies; but only low occupational position is mentioned once in the introduction with no consequences in the analysis and discussion. The Authors find almost no differences on psychosocial exposures regarding migrant status; perhaps because what would explain main differences could be social class / occupational grade or employment conditions?

Even considering that the unemployed at the time of interview were excluded in this study, the study population may still be a mixture of persons employed under very different conditions that are related to exposures and outcomes, mainly if the work is temporary or permanent, and full-time or part-time; low or high grade. The Authors mention that Australian-born workers have a much lower proportion (12 vs 25%) of workers with a postgraduate degree than migrants Non-ESC-born, but no differences in occupational class / category /grade; temporary / permanent, full / partial? I wonder if HILDA does not include any variable that could be used at least as a proxy of these constructs to take them into account in the analysis as a potential confounders, since they may relate to both exposures and outcome; but I think these are two important issues that should be addressed in the paper, and at least mentioned in the introduction and considered in the discussion–if no possible in the analysis

Response:

Thank you for the comments. 

We agree that many factors―not only socio-demographic but also employment-related―potentially confound the relationship between psychosocial job characteristics and mental health. We acknowledge this, and tested contract type, full/part-time and occupational skill level as potential confounders when we wrote the previous manuscript. However, these variables did not greatly impact on the beta coefficients in the models, and we have presented the most parsimonious models as our main results. Furthermore, it is arguable whether all such ‘potential confounders’ should be included in these models. Contract type and occupational skill level may be potential mediators of associations between educational attainment and psychosocial job characteristics, as well as mental health. To clarify this issue, we have developed and included a directed acyclic graph (DAG), and included it as supporting information S2 Fig in the revised manuscript. Thus, we would argue that controlling for contract type and occupational skill level in addition to educational attainment may result in an over-adjustment. Further, full/part-time overlaps with working hours and might be best operationalised as a work characteristic. 

Nevertheless, we have added sensitivity analyses to show results after controlling for these employment-related factors―including contract type, full/part-time and occupational skill level (see supporting information S1 Table), and explained why we did not include them in the main analysis in the Results section. 

Finally, we have used the most parsimonious model because interactions were tested in our study, and unnecessary covariates would further reduce statistical power (interaction tests are often under-powered).

The corresponding revisions to the text are: 

Sensitivity analysis (page 11, line 231-247)

Some employment-related factors―including contract type, full/part-time and occupational skill level―were potential confounders of the relationship between psychosocial job characteristics and mental health [17-19]. However, contract type and occupational skill level may be potential mediators of associations between educational attainment and psychosocial job characteristics, as well as mental health (see supporting information S2 Fig); thus, controlling them in addition to educational attainment may result in an over-adjustment. Furthermore, full/part-time overlaps with working hours and might be better treated as a job characteristic. We would like to include a minimal number of confounders because interactions were tested in this study―which is often under-powered, and unnecessary covariates would further reduce the statistical power. Therefore, we conducted sensitivity analyses to test whether these employment-related factors should be controlled as confounders by including them one by one into linear regressions, and then, simultaneously with gender, age and educational attainment (see supporting information S1 Table).

Contract type was a four-category variable: permanent contract, casual employment, fix-term contract and self-employment. Full-time workers were those who work ≥35 hours per week. Occupational skill level included four categories: highest skill, mid-high skill, mid-low skill and lowest skill [20].

Results (page 13-14, line 291-294)

Based on the results of sensitivity analysis (see supporting information S1 Table), including contract type, full/part-time and occupational skill level, either separately or simultaneously, into the linear regressions only resulted in small changes in the coefficients; therefore, they were excluded from the main analysis in this study.

S1 Table. Psychosocial job characteristics and mental health: Unadjusted and adjusted linear regression results (n=8917).

 Skill discretion Decision authority Job insecurity

Unadjusted Beta coefficient (95% CI) 0.40 (0.28, 0.52) *** 0.41 (0.34, 0.48) *** -1.04 (-1.13, -0.96) ***

 Constant 70.96 69.84 83.91

Adjusted for gender Beta coefficient (95% CI) 0.40 (0.28, 0.52) *** 0.39 (0.32, 0.46) *** -1.07 (-1.15, -0.98) ***

 Constant 72.20 71.14 85.57

Adjusted for age Beta coefficient (95% CI) 0.40 (0.28, 0.52) *** 0.39 (0.31, 0.46) *** -1.05 (-1.13, -0.97) ***

 Constant 69.70 69.31 82.30

Adjusted for educational attainment Beta coefficient (95% CI) 0.41 (0.29, 0.53) *** 0.41 (0.34, 0.49) *** -1.05 (-1.13, -0.96) ***

 Constant 70.87 69.80 83.51

Adjusted for contract type Beta coefficient (95% CI) 0.35 (0.23, 0.48) *** 0.42 (0.34, 0.49) *** -1.04 (-1.12, -0.96) ***

 Constant 71.77 70.29 83.80

Adjusted for full/part-time Beta coefficient (95% CI) 0.35 (0.22, 0.47) *** 0.39 (0.32, 0.46) *** -1.04 (-1.12, -0.95) ***

 Constant 72.10 70.63 84.48

Adjusted for occupational skill level Beta coefficient (95% CI) 0.41 (0.28, 0.54) *** 0.41 (0.34, 0.49) *** -1.05 (-1.13, -0.96) ***

 Constant 71.51 70.36 84.51

Adjusted﹫ Beta coefficient (95% CI) 0.38 (0.25, 0.51) *** 0.40 (0.32, 0.48) *** -1.08 (-1.17, -1.00) ***

 Constant 71.81 71.16 83.85

Results of univariable and multivariable linear regressions; Mental health was measured by MHI-5 score; *** p<0.001; ﹫Adjusted model adjusted for gender, age, educational attainment, contract type, full/part-time and occupational skill level simultaneously

Minor comments

R1-2. In my opinion, data source section of the article is quite difficult to read, with many population numbers given that may confuse the reader on the final sample size.

Response:

Thank you for the comment. We have revised this paragraph.

The corresponding revisions to the text are (page 6, line 123-133):

The data come from the Household Income and Labour Dynamics in Australia Survey (HILDA). HILDA is a nationally representative sample of Australian households, whose data collection began in 2001 and is conducted annually [21]. It covers a wide range of information on family life, household composition, income, labour market activity, employment, socioeconomic status and health. All household members aged 15 and older were interviewed through face-to-face or telephone interviews. In wave 1 (2001), 13969 persons responded to the survey, and the response rate was 92.3%. In 2011, an additional 4280 persons were added to top up the sample size to allow a better representation of the Australian population. Wave 14 (2014-2015) was used in the current study. In this wave, 17325 individuals responded to the survey, and the response rate for the parent sample was 80.8% [22]. The final analytic sample consists of 8969 respondents. More information on the analytic sample for this paper is described under ‘Inclusion and exclusion criteria’ below.

R1-3. What is the final response rate?

Response:

The final response for the parent sample rate was 80.8%. We have reported it in the Materials and methods section now. Please see the response to your previous comment.

R1-4. I would suggest edit Table 1, making it clear that columns 3 to 7 refer only to overseas-born workers.

Response:

Thank you for the comment. We have revised Table 1 accordingly.

The corresponding revisions to the text are (page 12, line266-267):

Table 1. Descriptive statistics on socio-demographic characteristics by migrant status (n=8969).

 Australian-born, n (%) Migrant workers

 Overseas-born, n (%) Main-ESC-born†, n (%) Non-ESC-born†, n (%) Arrived ≤5 years, n (%) Arrived 6-10 years, n (%) Arrived ≥11 years, n (%)

Overall 7238 (80.70) 1731 (19.30) 804 (8.96) 927 (10.33) 154 (1.72) 199 (2.22) 1378 (15.36)

Gender 

Male 3626 (50.10) 919 (53.09) 461 (57.34) 458 (49.41) 76 (49.35) 111 (55.78) 732 (53.12)

Female 3612 (49.90) 812 (46.91) 343 (42.66) 469 (50.59) 78 (50.65) 88 (44.22) 646 (46.88)

Age (years) 

15-24 1486 (20.53) 109 (6.30) 45 (5.60) 64 (6.90) 29 (18.83) 25 (12.56) 55 (3.99)

25-34 1662 (22.96) 370 (21.37) 134 (16.67) 236 (25.46) 84 (54.55) 81 (40.70) 205 (14.88)

35-44 1498 (20.70) 416 (24.03) 183 (22.76) 233 (25.13) 29 (18.83) 64 (32.16) 323 (23.44)

45-54 1582 (21.86) 493 (28.48) 269 (33.46) 224 (24.16) 8 (5.19) 25 (12.56) 460 (33.38)

55-64 1010 (13.95) 343 (19.82) 173 (21.52) 170 (18.34) 4 (2.60) 4 (2.01) 335 (24.31)

Educational attainment 

High school or lower 2621 (36.21) 398 (22.99) 218 (27.11) 180 (19.42) 28 (18.18) 38 (19.10) 332 (24.09)

Diploma or certificate 2530 (34.95) 553 (31.95) 302 (37.56) 251 (27.08) 38 (24.68) 44 (22.11) 471 (34.18)

Bachelor 1232 (17.02) 405 (23.40) 142 (17.66) 263 (28.37) 50 (32.47) 57 (28.64) 298 (21.63)

Postgraduate 855 (11.81) 375 (21.66) 142 (17.66) 233 (25.13) 38 (24.68) 60 (30.15) 277 (20.10)

† Main-ESC-born: Born in a Main-English-speaking country; Non-ESC-born: Born in a Non-English-speaking country

Reviewer 2

This manuscript assesses whether migrant workers are more vulnerable to job stressor exposures in terms of mental health, based on a survey in 2014-2015 among 8970 persons in an Australian working population-representative sample. Few previous studies have examined migrant status-based differences in job stressor—mental health associations. Therefore, whether migrants are differentially affected by job stressor exposures is unclear, especially when language proficiency and years since arrival in host country are taken into account. Thus, this topic is relevant and interesting.

Major comments

R2-1. The manuscript is mostly well-written. However I have some comments. The authors conclude that they found little evidence that the relationship between job stressor exposures and mental health is modified by migrant status. However, the stratified analyses, presented in the figures, show that migrants who arrived ≤5 years previously differed from other groups. Also migrants born in a Non-English speaking country seem to differ somewhat from other groups. The stratified analyses, thus, indicate that there is effect modification. This is my major comment.

Response

Thank you for the comments. In this study, the effect-measure modification was investigated by fitting interaction terms and we followed standard statistical procedures to determine whether there was potential effect-measure modification [23], and have now expanded our discussion of this below and in the manuscript.

Firstly, we tested whether the potential interaction/effect-measure modification was statistically significant using the Likelihood ratio (LR) test. This involved fitting a fully adjusted model-1 with variables A and B (A+B) and then a second fully adjusted model-2 with an interaction term for A and B (A + B + AxB). The LR test determined whether the interaction/effect-measure modification of A and B was statistically significant by a p-value. However, the p-value only provides a measure of statistical significance and no information on the magnitude of associations. Therefore, it was followed by a second step―graphing the potential effect-measure modification with 95% confidence intervals (CI) to visualize the magnitude of the association and see if there was any indication of possible effect-measure modification. The third step would be stratifying the association between exposure and outcome by the effect-measure modifier if the LR test result had a p-value less than 0.05.

In some of our analyses, the LR test resulted in a large p-value―indicating no statistical evidence of effect-measure modification, while graphs showed some suggestion of effect-measure modification. This could be due to the small sample size of migrants who arrived ≤5 years previously (n=154), thus limiting statistical power. That is, the small subgroup may have a wide 95% CI and, as a result, some 95% CIs were overlapped in our models and, thus, were consistent with the LR test results giving p-values far great than 0.05. We have revised the graphs of the psychosocial job characteristic―mental health associations by migrant status with 95% CIs (see Figs 2 and 3). It can be seen that the 95% CI of migrants who arrived ≤5 years largely overlapped with other subgroups even though the slopes of this subgroup looked different from that of other groups. This explains our reporting that there is in some instances a suggestion of effect-measure modification, though a larger study with higher power would be required to further investigate this.

We have revised the Materials and methods, Results, and Discussion sections to explain this clearly and have now included the subgroup 95% CI for each plotted line in our graphs.

The corresponding revisions to the text are:

Materials and methods (Page 10, line 220-228)

Third, Likelihood ratio (LR) tests were used to assess whether the potential effect-measure modification by migrant status of psychosocial job characteristic―mental health relationships were statistically significant. Considering that LR tests only provided measures of statistical significance without information on the relative magnitude of associations, they were complemented for descriptive purposes by graphing the relationships between psychosocial job characteristics and mental health by migrant status with 95% confidence intervals (CIs) to visualise the magnitude of associations and see if there was any indication of possible effect-measure modification. If the LR test results were significant (p<0.05), the relationships between psychosocial job characteristics and mental health would be stratified by the effect modifiers.

Results (Page 14, line 305-309)

When graphing the effect-measure modification results, we found that in both skill discretion―mental health (see Fig 2) and decision authority―mental health (see Fig 3) relationships, the regression lines for migrants who arrived ≤5 years previously appeared to differ from the lines for other groups. The slopes of the regression lines for migrants who arrived ≤5 years previously were close to zero; however, their 95% CIs largely overlapped with that of other groups.

Discussion 

(Page 15-16, line 334-342)

Inconsistent with our hypothesis, the magnitude of associations between psychosocial job characteristics and mental health did not vary by migrant status in our working Australian population sample despite investigating migrant status with three distinct measures. However, there was a suggestion that psychosocial job characteristic―mental health associations for migrants who had arrived in Australia ≤5 years previously differed from the associations for migrants who had been living in Australia for longer, though we could not rule out the possibility that this difference arose by chance. Skill discretion and decision authority were positively associated with MHI-5 score for both Australian-born workers and most subgroups of migrants, but there were non-statistically significant suggestions that regression lines were fairly flat for the most recently arrived migrants.

(Page 16-17, line 358-364)

We do, however, find suggestive, non-statistically significant evidence that the most recently arrived migrants may not be sensitive to skill discretion and decision authority, suggesting associations for these two psychosocial job characteristics are stronger in Australian-born workers (see Figs 2 and 3); however, these apparent differences may be due to the small sample size of the subgroup of migrants who arrived in Australia ≤5 years previously (n=154). Future study with a larger sample size would be required to resolve this question.

Minor comments

Other comments are mostly spelled out in detail for the respective sections of the manuscript.

Title:

R2-2. The title of the manuscript ("vary by migrant status") does not quite correspond with the aim of the study ("more vulnerable").

Response:

Thank you for the comment. This study aimed to test whether the vulnerability to psychosocial job characteristic-associated impacts on mental health varies by migrant status and we hypothesized migrant workers may be more vulnerable than native-born workers. We have revised our Title to make it clearer.

The corresponding revision to the text is:

Title

Do psychosocial job characteristic-associated impacts on mental health differ by migrant status? An Australian comparative analysis

Abstract:

R2-3. The year of the survey would be much more informative to the reader than the wave number. 

Response:

Thank you for the comment. We have added the year of the survey.

The corresponding revision to the text is:

(page 2, line 34-36)

……A total of 8969 persons from wave 14 (2014-2015) of the Household Income and Labour Dynamics in Australia Survey were included in the analysis…….

R2-4. In order to understand the results, the reader would need to know that higher MHI-5 scores indicate better mental health (contrary to Hopkins symptom check List, HSCL), and I suggest adding this information to the abstract.

Response:

Thank you for the comment. We have added the indication of MHI-5 in the Abstract.

The corresponding revision to the text is:

(page 2, line 37-38)

Mental health was assessed via a Mental Health Inventory-5 score (MHI-5), with a higher score indicating better mental health.

Introduction:

R2-5. Lines 98-101: The reference to this sentence (Migrant workers…) seems to be missing.

Response:

Thank you for the comment. We have added references.

The corresponding revision to the text is:

(page 5, line 106-109)

Migrant workers in Australia have been shown to experience lower skill discretion/complexity and higher job insecurity than Australian-born workers, especially those from Non-English speaking countries and recently arrived migrants [24, 25].

Material and methods:

R2-6. It is stated that the study participants were interviewed through face-to-face or telephone interviews. However, there also seems to have been a Self-Completion Questionnaire, mentioned further down.

Response:

Thank you for the comment. The HILDA survey comprised four different instruments: the Household Form (HF), the Household Questionnaire (HQ), the Person Questionnaire (PQ), and the Self-Completion Questionnaire (SCQ). The vast majority of data was collected through face-to-face interviews, and a small number was collected over the telephone. However, the SCQ data was not collected by face-to-face interview or telephone because the SCQ covers some topics that may make respondents slightly uncomfortable. All persons who completed the PQ were asked to complete the SCQ, but the SCQ was left with the respondents and collected at a later date or returned in the mail. We have revised our manuscript to only discuss the SCQ in the Missing data sub-section in the hope this may avoid further confusion. 

The corresponding revisions to the text are:

Materials and methods (page 6, line 126-133)

All household members aged 15 and older were interviewed through face-to-face or telephone interviews. In wave 1 (2001), 13969 persons responded to the survey, and the response rate was 92.3%. In 2011, an additional 4280 persons were added to top up the sample size to allow a better representation of the Australian population. Wave 14 (2014-2015) was used in the current study. In this wave, 17325 individuals responded to the survey, and the response rate for the parent sample was 80.8% [22]. The final analytic sample consists of 8969 respondents. More information on the analytic sample for this paper is described under ‘Inclusion and exclusion criteria’ below.

Missing data (page 9, line 198-201)

Based on the reason for missing data provided by HILDA, 1184 among the 1586 were due to non-response to the self-completion questionnaire (SCQ) of the HILDA survey, where psychosocial job characteristics were measured.

R2-7. P. 6: Job control was measured by two subscales (line 140), however, further down one of the subscales (decision authority) is referred to as (job) control (line 143). These were measured from 1 to 7, and seem to be a Likert scale, but this is not stated, as for the outcome and job insecurity.

Response:

We defined decision authority by ‘Decision authority assess the opportunity for control……’ (Page 7, line 150) there. We have explained all the three psychosocial job characteristic scales and MHI-5 were Likert scale in our manuscript. For skill discretion and decision authority, we said: “All the items were scored from 1 ‘strongly disagree’ to 7 ‘strongly agree’……” (Page 7, line 153). For job insecurity, we said, “Job insecurity……, which was assessed by three items with the same 7-point Likert scale” (Page 7, line 159). For the MHI-5, we said: “The MHI-5 consists of five six-point Likert items…… (page 6-7, line 138-139)”.

R2-8. P. 7: I would suggest referring to confounders and effect modifiers as POTENTIAL confounders and effect modifiers in the Methods section, before doing the analyses.

Response:

Thank you for the comments. We have revised the Materials and methods section using “Potential confounders” and “Potential effect modifier”. 

R2-9. P. 8: Inclusion, exclusion and missing data: I would suggest including a flow chart showing this. In the Missing data part, it is not so clear whether the different categories of missing data are overlapping or if the numbers with missing data for each step only include those without missing data in the previous step.

Response:

Thank you for the comments. We have revised the Inclusion and exclusion section and created a participant flow chart (see Fig 1) to make it clearer. There were 1606 missing cases (10575-8969=1606), the categories were not overlapping. It also has been shown in Fig 1.

The corresponding revisions to the text are (page 9, line 189-192):

The parent sample consisted of 17325 individuals in wave 14. As shown in Fig 1, 15231 of them were aged 15-64 years and of these, 10575 were employed, 10569 provided information of all confounders and migrant status. Among these 10569, 8983 observations answered all the items of the three psychosocial job characteristic scales, and 8969 also had MHI-5 score and, thus, were included in the analysis.

R2-10. Lines 191-195: The different reasons for refusing to answer (or not stated?) do not seem so interesting to me (but I may be wrong).

Response:

Thank you for the comments. A total of 1586 sample were excluded from our analysis due to missing items in psychosocial job characteristic scales, which may result in bias due to missing data. We described the missing data in detail to show that in most instances, all psychosocial job characteristic items were missing for all three scales; thus, possibilities for value substitution were limited. Now, as per your suggestion, we have shortened the whole paragraph on Missing data.

The corresponding revision to the text is (page 9-10, line 197-208):

There were 1606 respondents, accounting for 15.2% of the employed sample excluded from our analysis―among whom, 1586 were due to missing items of psychosocial job characteristic scale. Based on the reason for missing data provided by HILDA, 1184 among the 1586 were due to non-response to the self-completion questionnaire (SCQ) of the HILDA survey, where psychosocial job characteristics were measured. Other reasons included the SCQ not being asked (n=286), multiple responses to the SCQ (n=21) and refused/not stated (n=95). Among the 95 who refused to answer, 9 were missing all items for the three psychosocial job characteristic scales. Based on these reasons, a total of 1479 (1184+286+9) were missing all items of the three psychosocial job characteristic scales; moreover, a small number were excluded due to multiple responses (n=21) and missing information of education (n=1), migrant status (n=5) and MHI-5 score (n=14). This left 86 respondents who were missing one or two items of the psychosocial job characteristic scales thus being excluded from analysis, limiting possibilities for value substitution (e.g., mean substitution of single missing items).

R2-11. Lines 195-197: Were the 86 respondents excluded?

Response:

Yes, they were those who were excluded from the analysis due to missing one or two items in any of the psychosocial job characteristic scales. 

R2-12. P. 9, lines 200-202: As all p-values are <0.001, I suggest writing this only once, at the end of the sentence. The same comment applies to the presentation of the characteristics of subpopulations (P. 10, lines 228-233).

Response:

Thank you for the comments. We have revised the paragraph according to your comments.

The corresponding revisions to the text are:

Missing data (page 10, line 210-214)

Compared to the observations included in the analysis, the exclusions were more likely to be male (56.4% vs. 50.7%), younger (15_24 years: 24.3% vs. 17.8%), lower educated (high school or lower: 38.7% vs. 33.7%), lower in skill level (low skill level: 18.4% vs. 14.0%; high skill level: 32.6% vs. 38.7%), casual employment (23.8% vs. 19.0%) and Non-ESC-born migrants (13.8% vs. 10.3%) (all p-values <0.001).

Results (page 12, line 257-264)

The differences in gender, age and educational attainment between Australian-born workers and migrant workers were apparent (all p-values <0.001). For example, Australian-born workers included a significantly lower proportion of males than Main-ESC-born workers (50.1% vs. 57.3%). Australian-born workers included a significantly lower proportion of workers with a postgraduate degree compared to Non-ESC-born workers (11.8% vs. 25.1%), especially compared to migrants who arrived in Australia between 6 to 10 years previously (30.2%). In contrast, Australian-born workers included a significantly higher proportion of workers between 15 to 24 years of age than overseas-born workers (20.5% vs. 6.3%).

Statistical analysis:

R2-13. Linear regression was used to assess the relationships between job stressors and mental health. Did you check that these relationships actually are linear, by plotting the exposure and outcome data? If they are not, it might be more correct to treat the variables as categorical data, e.g. by combing the levels into a few categories, using the highest category (for job control) and the lowest category (for job insecurity) as reference, respectively.

Response:

Yes, we have checked the linear relationships between psychosocial job characteristics and MHI-5. Though they were not strong linear relationships (as shown in the example below), we still considered them linear and, hence, decided against creating categorical variables. From an epidemiological research perspective we were interested in the effect of continuous determinants on continuous outcomes and, thus, dichotomizing/categorizing a continuous variable may lead to a loss of information and statistical power. In this study, interactions were tested―which is often under-powered, therefore, we prefer to treat psychosocial job characteristics as continuous to maximise statistical power.

R2-14. LR tests were used to assess migrant status as an effect modifier of the job stressor—mental health relationships. However, this may be better evaluated in stratified analyses, as commented below (Results section).

Response:

Thank you for your comment. We have addressed it in our response to major comment R2-1.

Results

R2-15. Some of the results of Table 1 are presented in the text of this section, without the decimals, which is perfectly fine and, I would also suggest showing maximum 1 decimal in the table.

Response:

Thank you for your comments. We prefer to keep 2 decimals in the tables because our other papers kept 2 decimals and we would like to be consistent. Following your suggestion, we have revised our manuscript to present 1 decimal in the whole text.

R2-16. A couple of places in the Results section, the authors write that the results suggest (lines 245, 275) or indicate (l. 359) something. This is an interpretation of the results and should be moved to the discussion.

Response:

Thank you for your comments. We have revised the Results section and move them to the Discussion section.

The corresponding revisions to the text are: 

Results

(page 13, line 286-289)

……the difference in predicted MHI-5 score between the minimum and maximum job characteristic scale values were 71.8—76.6 for skill discretion, 71.1—78.5 for decision authority, and 80.8—62.1 for job insecurity. The range of predicted MHI-5 score of job insecurity was much wider than that of skill discretion and decision authority.

(page 14, line 308-312)

The slopes of the regression lines for migrants who arrived ≤5 years previously were close to zero; however, their 95% CIs largely overlapped with that of other groups. In the job insecurity―mental health relationships, the regression lines for all groups, including Australian-born workers and all migrant subgroups based on COB, dominant language of COB and years since arrival in Australia, were almost overlapping (see Fig 4).

Discussion (page 16, line 339-345)

Skill discretion and decision authority were positively associated with MHI-5 score for both Australian-born workers and most subgroups of migrants, but there were non-statistically significant suggestions that regression lines were fairly flat for the most recently arrived migrants. Job insecurity was negatively associated with the MHI-5 score for both Australian-born workers and all subgroups of migrant workers. Furthermore, job insecurity was a stronger determinant of mental health than skill discretion and decision authority in all workers, with both a larger magnitude of effect on MHI-5 score and a wider range of change in predicted MHI-5 score.

R2-17. The value of the constant is shown in Table 2, but not mentioned in the Results or Discussion sections, as far as I can see. Do the differences in the constant value indicate important differences between the subgroups?

Response:

The regression constant represents the predicted MHI-5 score when the measured psychosocial exposure=0. However, because our focus in this study is the relationship between psychosocial job characteristics and mental health, we focused more on beta coefficients.

We have made use of the constant values in one small way in the Results section. The predicted MHI-5 score in the Results section was calculated by: the constant value + beta coefficient x psychosocial job characteristic score from minimum to maximum in the unadjusted model (e.g. for skill discretion: 71.01+0.40x2=71.81, 71.01+0.40x14=76.61). 

Results (Page 13, line 284-288)

Moreover, combined with the constant values (since gender, age and educational attainment were not confounders for psychosocial job characteristic―mental health relationships, here, we used the unadjusted model to make the analysis easy), the difference in predicted MHI-5 score between the minimum and maximum job characteristic scale values were 71.8—76.6 for skill discretion, 71.1—78.5 for decision authority, and 80.8—62.1 for job insecurity.

Discussion (Page 16, line 343-345)

Furthermore, job insecurity was a stronger determinant of mental health than skill discretion and decision authority in all workers, with both a larger magnitude of effect on MHI-5 score and a wider range of change in predicted MHI-5 score.

R2-18. I would suggest writing what kind of analyses the results of the table are based on.

Response:

Thank you for this comment. We have revised the table title and added “Results of univariable and multivariable linear regressions” in the footnote of Table 2. Furthermore, according to your comment, we have revised figure legends to include the analysis methods as well. Please see our response to your major comment (R2-1).

The corresponding revision to the text is (Page 13-14, line 292-295):

Table 2. Psychosocial job characteristics and mental health: Unadjusted and adjusted linear regression results (n=8969).

Psychosocial job characteristics Unadjusted Adjusted for gender Adjusted for age Adjusted for education Adjusted﹫

 Cons Coef. (95%CI) Cons Coef. (95%CI) Cons Coef. (95%CI) Cons Coef. (95%CI) Cons Coef. (95%CI)

Skill discretion 71.01 0.40 (0.28, 0.52) *** 72.25 0.39 (0.27, 0.51) *** 69.78 0.40 (0.27, 0.52) *** 70.93 0.41 (0.28, 0.53) *** 71.02 0.40 (0.28, 0.53) ***

Decision authority 69.91 0.41 (0.33, 0.48) *** 71.20 0.38 (0.31, 0.45) *** 69.42 0.38 (0.31, 0.45) *** 69.88 0.41 (0.33, 0.48) *** 70.71 0.36 (0.29, 0.44) ***

Job insecurity 83.92 -1.04 (-1.13, -0.96) *** 85.57 -1.07 (-1.15, -0.98) *** 82.36 -1.05 (-1.13, -0.97) *** 83.54 -1.05 (-1.13, -0.97) *** 83.97 -1.08 (-1.16, -0.99) ***

Results of univariable and multivariable linear regressions; Mental health was measured by MHI-5 score; Cons was the constant value; Coef. was the beta coefficient; *** p<0.001; ﹫Adjusted model adjusted for gender, age and educational attainment simultaneously

R2-19. Based on the LR test results, the authors state that there was no statistical evidence confirming migrant status acting as effect modifiers of the job stressor exposure—mental health relationships. This is also the conclusion of the manuscript. However, the stratified analyses, presented in the figures, show that migrants who arrived ≤5 years previously differed from other groups. Also migrants born in a Non-English speaking country seem to differ somewhat from other groups. The stratified analyses, thus, indicate that there is effect modification. To me, all three job stressors seem to have less effect on mental health among migrants who arrived ≤5 years previously, compared to other groups. They also seem to be less vulnerable to high job insecurity. The lines of the Australian-born group generally seem to lie somewhat higher than the lines of other groups for the two job control subscales. Could this indicate an interesting result?

Response:

Thank you for the comments. We have addressed the first part of the comments in our response to major comment R2-1 above. In response to the final point:

The regression lines of Australian-born workers were higher than that of migrant workers in skill discretion/decision authority―mental health associations, which indicate when the skill discretion or decision authority was at the same level, Australian-born workers may have a higher MHI-5 score than migrant workers. However, LR test results were non-significant in our study, which indicate the differences in these regression lines may have arisen by chance. In Figs 2 and 3, it showed that the 95% CIs of the regression lines were overlapping.

R2-20. I am not a statistician, and will not speculate why the LR test did not indicate effect modification. However, I will just mention that results of effect modification analyses will depend on the measures used, therefore, Rothman prefer to use the term "effect MEASURE modification" (e.g. in the book Modern Epidemiology), indicating that you e.g. may find effect modification on an additive scale, but not on a relative scale, or vice versa. Is it correct to test effect modification in linear analysis by adding a multiplicative interaction term? Or should this preferably be tested in other ways?

Response:

Thank you for your comments. For the LR test comment, we have addressed it in our response to major comment R2-1.

Yes, we agree with you it should be ‘effect-measure modification’. Our study is indeed a salient demonstration of the need for this term―because we measured the single construct of migrant status in three distinct ways. We have changed ‘effect modification’ to ‘effect-measure modification’ throughout the manuscript. 

For the additivity/multiplicativity question: testing interaction with a product term in linear regression (as we have done) tests departure from additivity, however, in logistic regression it refers to interaction as departure from multiplicativity [26].

Discussion

R2-21. The first sentence (lines 294-296) is not according to the results, as far as I can see, as mentioned above.

Response:

We have addressed this comment in our response to major comment R2-1. This sentence is the conclusion of what we draw from our results. 

R2-22. The authors refer to a study by Hoppe indicating that migrants from some countries may experience that working conditions in the new country is much better (lines 327-328). The opposite may also be true, if the migrants' education is not approved in the new country, and they have to accept manual work with lower job control and higher job insecurity.

Response:

Yes, we agree with you that migrants may be exposed to higher levels of adverse psychosocial job characteristics if their qualifications are not accepted in the host country and they have to do jobs with poor working conditions. But here we cited Hoppe to explain a possible reason why migrant workers are not more vulnerable than native-born workers even though they experienced higher levels of psychosocial exposures―comparing with their former working conditions, the working conditions now are better even it is worse than that of native-born workers―rather than why they experienced higher levels of exposures.

R2-23. Page 16 lines 359-361: The authors state: "…apart from reducing occupational health inequalities of migrant workers, reducing job stressor exposures should improve mental health of Australia-born workers as well." This could easily result in occupational health inequalities being maintained.

Response:

Yes, we agree that reducing adverse psychosocial exposures equally for all workers would preserve occupational mental health inequalities between migrant and native-born workers; however, the job characteristic-associated mental health burden would be reduced. Moreover, the recommendations we highlighted in our study were to reduce adverse psychosocial job characteristic exposures for migrants in particular by improving migrants’ English skill, which would be specifically helpful to reduce adverse psychosocial job characteristic exposures among migrant workers and, thus, reduce occupational mental health inequalities between migrant and native-born workers.

R2-24. Lines 366-368: Maybe add that poor working conditions may lead to poor health, and therefore healthy worker survivor effect (there seemed to be a missing link).

Response:

Thank you for your comment. We have revised this limitation.

The corresponding revision to the text is (Page 19, line 409-411):

Second, our results may have been biased toward the null due to the healthy worker effect―some workers may have dropped out of the workforce due to poor working conditions thus underestimating exposure-outcome associations.

R2-25. Page 17, lines 373-375: Common method bias could also be a limitation of a study based on subjective/self-reported job stressors and mental health.

Response:

Thank you for your comment. We have revised this limitation.

The corresponding revision to the text is (Page 19, line 417-420):

Fourth, one limitation of our study is the reliance on self-report measures, both for psychosocial job characteristics and mental health, thus common method bias could potentially bias the relationship between psychosocial job characteristics and mental health.

References

R2-26. When referring to more than one authors in the text, the authors include two of the names, instead of using one name + et al. Is this according to the PLOS? For some references in the reference list, the journal titles are spelled out, while for others, the usual abbreviated forms are used. These should also be according to the style of the journal.

Response:

Thank you for your comments. Yes, we have used the Endnote style of PloS One to insert the references in our previous manuscript. Now, we have revised the reference list.

Language

R2-27. The language is generally good, but some sentences starting with "there is/were…" seem a bit incomplete (e.g. lines 101, 120, 122 and 125). A couple of places there is also an S missing in 3rd person present tense (lines 105 and 143). There are also a few other things. Line 365: Should "casual" be "causal"?

Response:

Thank you for your comments. We have edited and proofread the manuscript carefully.

References

1. Landsbergis PA, Grzywacz JG, LaMontagne AD. Work organization, job insecurity, and occupational health disparities. Am J Ind Med. 2014;57(5):495-515. doi: 10.1002/ajim.22126. PubMed PMID: 23074099.

2. Centres for Disease Control and Prevention. Occupational health equity 2019 [updated 18 December 2019; cited 2020 18 February]. Available from: https://www.cdc.gov/niosh/programs/ohe/default.html.

3. LaMontagne AD. Commentary: Precarious employment: Adding a health inequalities perspective. J Public Health Policy. 2010;31(3):312-7. doi: 10.1057/jphp.2010.25. PubMed PMID: 20805803.

4. Abubakar I, Aldridge RW, Devakumar D, Orcutt M, Burns R, Barreto ML, et al. The UCL–Lancet Commission on Migration and Health: The health of a world on the move. Lancet. 2018;392(10164):2606-54. doi: 10.1016/S0140-6736(18)32114-7. PubMed PMID: 30528486.

5. Quandt SA, Arcury-Quandt AE, Lawlor EJ, Carrillo L, Marín AJ, Grzywacz JG, et al. 3-D jobs and health disparities: The health implications of Latino chicken catchers' working conditions. Am J Ind Med. 2013;56(2):206-15. doi: 10.1002/ajim.22072. PubMed PMID: 22618638.

6. Thamrin Y. A literature review of migrant workers’ health and safety. Jurnal Kesehatan Masyarakat Maritim. 2018;2(1):245-94.

7. Hargreaves S, Rustage K, Nellums LB, McAlpine A, Pocock N, Devakumar D, et al. Occupational health outcomes among international migrant workers: A systematic review and meta-analysis. The Lancet Global Health. 2019;7(7):e872-e82. doi: 10.1016/S2214-109X(19)30204-9. PubMed PMID: 31122905.

8. Sterud T, Tynes T, Mehlum IS, Veiersted K, Bergbom B, Airila A, et al. A systematic review of working conditions and occupational health among immigrants in Europe and Canada. BMC Public Health. 2018;18(1):770. doi: 10.1186/s12889-018-5703-3. PubMed PMID: 29925349.

9. WHO. Mental health in the workplace 2019 [updated May 2019; cited 2020 1 February ]. Available from: https://www.who.int/mental_health/in_the_workplace/en/.

10. OECD. Fit mind, fit job: From evidence to practice in mental health and work Paris: OECD Publishing; 2015.

11. Stansfeld S, Candy B. Psychosocial work environment and mental health—a meta-analytic review. Scand J Work Environ Health. 2006;32(6):443-62.

12. Llosa JA, Menéndez-Espina S, Agulló-Tomás E, Rodríguez-Suárez J. Job insecurity and mental health: A meta-analytical review of the consequences of precarious work in clinical disorders. Anales de Psicología. 2018;34(2):211-23. doi: 10.6018/analesps.34.2.281651.

13. Saijo Y, Yoshioka E, Kawanishi Y, Nakagi Y, Itoh T, Sugioka Y, et al. Effects of work burden, job strain and support on depressive symptoms and burnout among Japanese physicians. Int J Occup Med Environ Health. 2014;27(6):980-92. doi: 10.2478/s13382-014-0324-2. PubMed PMID: edselc.2-52.0-84916912162.

14. Theorell T, Hammarström A, Aronsson G, Träskman Bendz L, Grape T, Hogstedt C, et al. A systematic review including meta-analysis of work environment and depressive symptoms. BMC Public Health. 2015;15:738. doi: 10.1186/s12889-015-1954-4. PubMed PMID: 26232123.

15. Milner A, Witt K, LaMontagne AD, Niedhammer I. Psychosocial job stressors and suicidality: A meta-analysis and systematic review. Occup Environ Med. 2018;75(4):245-53. doi: 10.1136/oemed-2017-104531. PubMed PMID: 28851757.

16. Milner A, LaMontagne AD, Spittal MJ, Pirkis J, Currier D. Job stressors and employment precarity as risks for thoughts about suicide: An Australian study using the Ten to Men Cohort. Ann Work Expo Health. 2018;62(5):583-90. doi: 10.1093/annweh/wxy024. PubMed PMID: 29635407.

17. Milner A, Smith P, LaMontagne AD. Working hours and mental health in Australia: Evidence from an Australian population-based cohort, 2001–2012. Occup Environ Med. 2015;72(8):573-9. doi: 10.1136/oemed-2014-102791. PubMed PMID: 26101295.

18. LaMontagne AD, Milner A, Krnjacki L, Kavanagh AM, Blakely TA, Bentley R. Employment arrangements and mental health in a cohort of working Australians: Are transitions from permanent to temporary employment associated with changes in mental health? Am J Epidemiol. 2014;179(12):1467-76. doi: 10.1093/aje/kwu093. PubMed PMID: 103963370.

19. LaMontagne AD, Krnjacki L, Kavanagh AM, Bentley R. Psychosocial working conditions in a representative sample of working Australians 2001-2008: An analysis of changes in inequalities over time. Occup Environ Med. 2013;70(9):639-47. doi: 10.1136/oemed-2012-101171. PubMed PMID: 23723298.

20. Australian Bureau of Statistics. ANZSCO - Australian and New Zealand Standard Classification of Occupations, First Edition. Canberra, Australia: ABS, 2006.

21. Watson N, Wooden M. The Household, Income and Labour Dynamics in Australia (HILDA) survey: Wave 1 survey methodology. Melbourne: Melbourne Institute of Applied Economic and Social Research, University of Melbourne, 2002.

22. Summerfield M, Freidin S, Hahn M, La N, Li N, Macalalad N, et al. HILDA user manual-Release 15. Melbourne: Melbourne Institute of Applied Economic and Social ResearchResearch, University of Melbourne, 2016.

23. Hayes AF. Introduction to mediation, moderation, and conditional process analysis: A regression-based approach. Second ed. New York: Guilford publications; 2018. 223-66 p.

24. Liu X, Bowe SJ, Milner A, Li L, Too LS, Lamontagne AD. Differential exposure to job stressors: A comparative analysis between migrant and Australia-born workers. Ann Work Expo Health. 2019;63(9):975–89. doi: 10.1093/annweh/wxz073. PubMed PMID: 31621876.

25. Liu X, Bowe SJ, Milner A, Li L, Too LS, LaMontagne AD. Job insecurity: A comparative analysis between migrant and native workers in Australia. Int J Environ Res Public Health. 2019;16(21):4159. doi: 10.3390/ijerph16214159. PubMed PMID: 31661926.

26. Knol MJ, van der Tweel I, Grobbee DE, Numans ME, Geerlings MI. Estimating interaction on an additive scale between continuous determinants in a logistic regression model. Int J Epidemiol. 2007;36(5):1111-8. doi: 10.1093/ije/dym157.

---

## [Decision Letter · Decision Letter 1]

20 Jul 2020

PONE-D-19-35389R1

Do psychosocial job characteristic-associated impacts on mental health differ by migrant status? An Australian comparative analysis

PLOS ONE

Dear Dr. LIU,

Thank you for submitting your manuscript to PLOS ONE and for attending to many of the Reviewers' and my comments. As you will see, Reviewer 1 acknowledges that their concerns were successfully addressed. However, Reviewer 2 asks for more clarification with respect to the data analysis in general and the method to test migrant as an effect modifier in particular. I suggest to present in additional analyses the robustness and non-significance of the interaction effect. The additional analyses do not have to be part of the manuscript; a presentation in the response letter should suffice.

Furthermore, it was mentioned in the response letter that you adopted the job demand-control (JDC) model as theoretical framework for the study. I neither saw this clearly implemented in the manuscript nor do I believe that the JDC model would be enough because migrant status cannot be conceptualised as effect modifier by means of this model (alone). 

We invite you to submit a revised version of the manuscript that addresses the points raised during the review process.

A response letter that responds to each point raised by the academic editor and reviewer(s). You should upload this letter as a separate file labeled 'Response to Reviewers'.A marked-up copy of your manuscript that highlights changes made to the original version. You should upload this as a separate file labeled 'Revised Manuscript with Track Changes'.An unmarked version of your revised paper without tracked changes. You should upload this as a separate file labeled 'Manuscript'.

We look forward to receiving your revised manuscript.

Kind regards,

Dana Unger

Academic Editor

PLOS ONE

Reviewers' comments:

Reviewer's Responses to Questions

**Comments to the Author**

1. If the authors have adequately addressed your comments raised in a previous round of review and you feel that this manuscript is now acceptable for publication, you may indicate that here to bypass the “Comments to the Author” section, enter your conflict of interest statement in the “Confidential to Editor” section, and submit your "Accept" recommendation.

Reviewer #1: All comments have been addressed

Reviewer #2: (No Response)

2. Is the manuscript technically sound, and do the data support the conclusions?

Reviewer #1: Yes

Reviewer #2: Partly

3. Has the statistical analysis been performed appropriately and rigorously? 

Reviewer #1: Yes

Reviewer #2: No

4. Have the authors made all data underlying the findings in their manuscript fully available?

Reviewer #1: Yes

Reviewer #2: Yes

5. Is the manuscript presented in an intelligible fashion and written in standard English?

Reviewer #1: Yes

Reviewer #2: Yes

6. Review Comments to the Author

Reviewer #1: Congratulations for a good and relevant article and thank you for considering all my comments, hope they helped you.

Reviewer #2: I have reviewed this manuscript previously. The authors have responded adequately to most of my comments, and they have also addressed my major concern. However, I am still not convinced that the conclusion is correct.

In fig. 2 (Skill discretion), to the right, the change for Australian-born workers is from a score of approx. 72 for low Skill discretion, to a score of approx. 76 for high Skill discretion, i.e. a score change of approx. 4. For migrants with ≤ 5 years since arrival, the score change is only about 1 (73 to 74), while for migrants > 5 years since arrival, the difference is larger than for Australian-born workers. I can hardly believe that the difference between Australian-born (score change 4) and migrants with ≤ 5 years since arrival (score change 1) is not statistically significant, even though the latter group is small. The figure shows very different curves for the two groups.

The authors write (lines 329-330): Since the LR tests were not significant, the relationships between psychosocial job characteristics and mental health were not stratified by migrant status.

Is the performed LR test the best way of evaluating effect measure modification? Do stratified analysis confirm the results of the LR test?

In this LR test, the interaction term consists of a psychosocial job characteristic variable, which is continuous, and a migrant status variable, which is categorical with 4 categories. Is the migrant status variable treated correctly in the interaction analysis? Often the variables in an interaction term are either continuous or dichotomous (as mentioned in ref. 26 in the response to the reviewers). How does the analysis treat the migrant status variable? As a categorical variable (with no intrinsic ordering to the categories)? As an ordinal variable (with clear ordering of the categories)? As a numerical variable (with equal intervals between the values of the variable)? Or as a continuous variable? Are all these alternatives possible in an LR analysis, e.g. categorical variables? Are all assumptions met for this analysis?

What do the partly overlapping confidence intervals indicate? If the constant is very similar but the slope very different, they would probably partly overlap, but does it mean that they are not significantly different?

If I understand correctly, Australian-born workers are the reference group that different migrant groups are compared to. However, this is not always clear.

TITLE:

The title of the manuscript has been changed, however, I now find it difficult to read. The term "psychosocial job characteristic-associated impacts" describes the outcome, but is very long and complicated.

INTRODUCTION:

Lines 52-57 (and elsewhere): The term "occupational health inequalities" is used to describe one or specific groups in this paragraph, without referring to which groups they are compared to. Inequalities/differences - compared to which groups?

Line 75: Some of the psychosocial job characteristics are referred to as high or low, but this could also apply to job strain and job insecurity.

Lines 116-118: "We hypothesise that the psychosocial job characteristic―mental health relationships are stronger for migrants, especially … migrants recently arrived in Australia."

Why do you hypothesise this? Isn't it contrary to studies you refer to?

MATERIAL AND METHODS:

Line 191: "8983 observations answered…" Can an observation answer?

Lines 204-206: I suggest moving this sentence ("moreover, a small sample…") to the end of the paragraph, so that the last sentence ("This left 86 respondents…", lines 206-208) comes closer to the sentence "Among the 95 who refused to answer, 9 were missing all items…" (line 202), i.e. the 86 referred to in the current last sentence.

RESULTS:

Lines 256-264: I suggest reversing the comparisons (if Australian-born workers are the reference). E.g., instead of "Australian-born workers included a significantly lower proportion of males than Main-ESC-born workers", I suggest: "Main-ESC-born workers included a significantly higher proportion of males than Australian-born workers".

In order to better understand and interpret the results, I would recommend adding a table showing the distribution of exposures and outcome (e.g. mean and range), according to migrant categories.

Lines 284-285: "since gender, age and educational attainment were not confounders for psychosocial job characteristic―mental health relationships".

According to the model (S2 Fig), they were confounders, but they barely confounded the relationships, according to the results, i.e. they were only weak confounders.

DISCUSSION:

Lines 359-361: "… evidence that the most recently arrived migrants may not be sensitive to skill discretion and decision authority, suggesting associations for these two psychosocial job characteristics are stronger in Australian-born workers". You have not actually compared the two groups statistically, which I would recommend that you do.

Lines 366-367: "Various explanations as to why migrants would not have a higher vulnerability to psychosocial job characteristics are plausible." You mention "healthy immigrant effect" and some other possible explanations. It is also worth mentioning that demographics of the most recently arrived migrants appear to be quite different, compared to the Australian-born workers: 55 % are 25-34 years of age (23 % of Australian-born), and 57 % are bachelor or postgraduates (29 % of Australian-born). Although you have adjusted for age and education level, these differences may also indicate other differences, which are not adjusted for. "Healthy immigrant effect" may partly cover this.

Lines 390-395: The sentence is very long and difficult to understand.

The statement: "Since migrants and Australian-born workers have similar sensitivity to psychosocial job characteristics in terms of mental health" is very categorical and does not quite harmonise with the results, in my opinion.

Lines 417-420: In what direction would common method bias probably affect the results?

Usually this type of bias leads to stronger associations, which could be added.

Lines 420-424: This could also make it difficult to observe differences between migrants from Non-English-speaking countries and English-speaking countries.

REFERENCES:

Check Reference 49 (Laura).

7. PLOS authors have the option to publish the peer review history of their article (what does this mean?). If published, this will include your full peer review and any attached files.

Reviewer #1: No

Reviewer #2: **Yes: **Ingrid Sivesind Mehlum

---

## [Author Response · Author response to Decision Letter 1]

26 Aug 2020

Academic Editor

AE-1. Reviewer 2 asks for more clarification with respect to the data analysis in general and the method to test migrant as an effect modifier in particular. I suggest to present in additional analyses the robustness and non-significance of the interaction effect. The additional analyses do not have to be part of the manuscript; a presentation in the response letter should suffice. 

Response:

Thank you for your suggestion. We have added a supplemental table to report slopes of regression lines for the figures in this revised manuscript. We also reported the differences in slopes between migrant subgroups and Australian-born workers to Reviewer 2. Please see the response to R2-1. If the reviewer remains unsatisfied with our analytic approach to effect-measure modification, we would suggest a Statistical Reviewer. We stand by our analysis as appropriate, and our interpretation as correct. It is indeed an unusual situation for the authors to be arguing for a more conservative interpretation than a Reviewer(!).

The corresponding revisions to the text are:

1) Results section (page 15, line 318-319):

……The slopes of the regression lines in Figs 2-4 were reported in the supporting information S3 Table.

2) Supporting information S3 Table:

S3 Table. The slopes of regression lines for relationships between psychosocial job characteristics and mental health: stratified by migrant status and controlled for gender, age and educational attainment

Psychosocial job characteristics 

Migrant status Predicted mental health

 Coefficient SE P value 95% CI

Skill discretion Australian-born 0.38 0.07 0.00 0.24; 0.51

 Overseas-born 0.48 0.14 0.001 0.20; 0.75

 Main-ESC-born† 0.37 0.21 0.08 -0.04; 0.78

 Non-ESC-born† 0.55 0.19 0.003 0.19; 0.92

 Arrived ≤5 years 0.10 0.44 0.827 -0.76; 0.95

 Arrived 6-10 years 0.58 0.38 0.125 -0.16; 1.34

 Arrived ≥11 years 0.51 0.16 0.001 0.20; 0.83

Decision authority Australian-born 0.36 0.04 0.00 0.28; 0.44

 Overseas-born 0.37 0.08 0.00 0.21; 0.53

 Main-ESC-born† 0.27 0.12 0.03 0.03; 0.50

 Non-ESC-born† 0.47 0.12 0.00 0.24; 0.70

 Arrived ≤5 years -0.04 0.35 0.90 -0.73; 0.65

 Arrived 6-10 years 0.58 0.25 0.022 0.08; 1.08

 Arrived ≥11 years 0.37 0.09 0.00 0.19; 0.55

Job insecurity Australian-born -1.08 0.05 0.00 -1.18; -1.00

 Overseas-born -1.03 0.09 0.00 -1.21; -0.85

 Main-ESC-born† -0.99 0.14 0.00 -1.26; -0.99

 Non-ESC-born† -1.07 0.12 0.00 -1.31; -0.83

 Arrived ≤5 years -0.79 0.33 0.02 -1.44; -0.14

 Arrived 6-10 years -1.27 0.27 0.00 -1.80; -0.74

 Arrived ≥11 years -1.03 0.10 0.00 -1.24; -0.83

† Main-ESC-born: Born in Main-English-speaking country, Non-ESC-born: Born in Non-English-speaking country; Mental health was measured by MHI-5 score

AE-2. It was mentioned in the response letter that you adopted the job demand-control (JDC) model as theoretical framework for the study. I neither saw this clearly implemented in the manuscript nor do I believe that the JDC model would be enough because migrant status cannot be conceptualised as effect modifier by means of this model (alone).

Response:

Thank you for your comments. We adopted the Job Demand-Control (JDC) model to define the psychosocial job characteristics; however, we conceptualised migrant status as a potential determinant of adverse occupational exposures based on the framework developed by Landsbergis et al. (1) (Page 3, line 56-59; Supporting information S1 Fig), through which migrant status may act as an effect modifier of the relationship between psychosocial job characteristics and mental health. We have now explained this further in the Methods section.

The corresponding revision to the text is (page 7, line154-156):

Psychosocial job characteristics were the exposure variables, which were assessed based on the Job Demand-Control model [2] and skill discretion, decision authority and job insecurity were measured in this study.

Reviewer 2

R2-1. I have reviewed this manuscript previously. The authors have responded adequately to most of my comments, and they have also addressed my major concern. However, I am still not convinced that the conclusion is correct.

Response:

We do agree with the reviewer that the figures (non-statistical evidence) suggested there may be effect-measure modification; however, the statistical evidence and the largely overlapping 95% CI of the slopes of regression lines did not support that finding. We appreciate the reviewer’s advocacy for this finding, but we do not want to overstate. We do, however, believe the non-statistical evidence (differences in the slopes) suggests that our research questions were not fully answered at this stage, and that future studies with larger sample sizes and higher statistical power would serve to answer the question more definitively. This was the reason why we kept these figures in this manuscript even though the statistical evidence suggested no effect modification. Furthermore, it is important to note that testing effect modification is different from stratification. 

To further substantiate our finding that we do not have statistical confirmation of effect-measure modification, the slopes with 95% CI of regression lines of our figures are presented in supporting information S3 Table. Here, we also reported the differences in the slopes between migrant subgroups and Australian-born workers (see Table 1). It showed that the 95% CIs of the slopes for the migrants ≤5 years since arrival were so wide that they almost completely overlapped with those of the Australian-born workers. This indicated that these slopes are not statistically different from each other. This is likely due to the relatively small sample size of migrants ≤5 years since arrival (n=154) compared to the sample size of Australian-born workers (n=7238). None of the differences in the slopes between migrant subgroups and Australian-born workers were statistically different from zero, which has been supported by the LR test results.

Table 1. Relationships between psychosocial job characteristics and mental health stratified by migrant status and the differences in the slopes between each migrant subgroup and Australian-born workers (controlled for gender, age and educational attainment)

Psychosocial job characteristics 

Migrant status Predicted mental health

 Coefficient SE P value 95% CI

Skill discretion The slope of each subgroup Australian-born 0.38 0.07 0.00 0.24; 0.51

 Overseas-born 0.48 0.14 0.001 0.20; 0.75

 Main-ESC-born† 0.37 0.21 0.08 -0.04; 0.78

 Non-ESC-born† 0.55 0.19 0.003 0.19; 0.92

 Arrived ≤5 years 0.10 0.44 0.827 -0.76; 0.95

 Arrived 6-10 years 0.58 0.38 0.125 -0.16; 1.34

 Arrived ≥11 years 0.51 0.16 0.001 0.20; 0.83

 Differences in the slopes Overseas-born - Australian-born 0.10 0.16 0.00 -0.20; 0.41

 Main-ESC-born† - Australian-born -0.01 0.22 0.98 -0.44; 0.43

 Non-ESC-born† - Australian-born 0.18 0.20 0.38 -0.21; 0.57

 Arrived ≤5 years - Australian-born -0.28 0.44 0.52 -1.14; 0.58

 Arrived 6-10 years - Australian-born 0.21 0.39 0.60 -0.56; 0.97

 Arrived ≥11 years - Australian-born 0.13 0.17 0.45 -0.21; 0.47

Decision authority The slope of each subgroup Australian-born 0.36 0.04 0.00 0.28; 0.44

 Overseas-born 0.37 0.08 0.00 0.21; 0.53

 Main-ESC-born† 0.27 0.12 0.03 0.03; 0.50

 Non-ESC-born† 0.47 0.12 0.00 0.24; 0.70

 Arrived ≤5 years -0.04 0.35 0.90 -0.73; 0.65

 Arrived 6-10 years 0.58 0.25 0.022 0.08; 1.08

 Arrived ≥11 years 0.37 0.09 0.00 0.19; 0.55

 Differences in the slopes Overseas-born - Australian-born 0.01 0.09 0.90 -0.17; 0.19

 Main-ESC-born† - Australian-born -0.09 0.13 0.45 -0.34; 0.15

 Non-ESC-born† - Australian-born 0.11 0.12 0.37 -0.13; 0.36

 Arrived ≤5 years - Australian-born -0.40 0.35 0.26 -1.09; 0.29

 Arrived 6-10 years - Australian-born 0.22 0.26 0.39 -0.28; 0.73

 Arrived ≥11 years - Australian-born 0.01 0.10 0.91 -0.18; 0.21

Job insecurity The slope of each subgroup Australian-born -1.08 0.05 0.00 -1.18; -1.00

 Overseas-born -1.03 0.09 0.00 -1.21; -0.85

 Main-ESC-born† -0.99 0.14 0.00 -1.26; -0.99

 Non-ESC-born† -1.07 0.12 0.00 -1.31; -0.83

 Arrived ≤5 years -0.79 0.33 0.02 -1.44; -0.14

 Arrived 6-10 years -1.27 0.27 0.00 -1.80; -0.74

 Arrived ≥11 years -1.03 0.10 0.00 -1.24; -0.83

 Differences in the slopes Overseas-born - Australian-born 0.05 0.10 0.64 -0.15; 0.25

 Main-ESC-born† - Australian-born 0.10 0.15 0.51 -0.19; 0.39

 Non-ESC-born† - Australian-born 0.01 0.13 0.93 -0.25; 0.27

 Arrived ≤5 years - Australian-born 0.29 0.34 0.387 -0.37; 0.95

 Arrived 6-10 years - Australian-born -0.19 0.27 0.49 -0.72; 0.35

 Arrived ≥11 years - Australian-born 0.05 0.11 0.66 -0.17; 0.27

† Main-ESC-born: Born in Main-English-speaking country, Non-ESC-born: Born in Non-English-speaking country; Mental health was measured by MHI-5 score

R2-2. In fig. 2 (Skill discretion), to the right, the change for Australian-born workers is from a score of approx. 72 for low Skill discretion, to a score of approx. 76 for high Skill discretion, i.e. a score change of approx. 4. For migrants with ≤ 5 years since arrival, the score change is only about 1 (73 to 74), while for migrants > 5 years since arrival, the difference is larger than for Australian-born workers. I can hardly believe that the difference between Australian-born (score change 4) and migrants with ≤ 5 years since arrival (score change 1) is not statistically significant, even though the latter group is small. The figure shows very different curves for the two groups.

Response:

Statistical testing effect modification of migrant status is to assess whether the differences in the associations between psychosocial job characteristics and mental health (slopes of regression lines)―rather than the differences in the changes in predicted MHI-5 scores with psychosocial job characteristics changing from the minimum to the maximum―between migrant subgroups and Australian-born workers are significantly different from zero. As we responded in comment R2-1, none of the differences in slopes were significantly different from zero.

We showed the three graphs side by side in our paper to facilitate comparison of the results for the three different measures of migrant status used. The combining of the three figures compressed the x-axis, and the y-axis had a scale from 60 to 80―which makes the differences in slopes clearly, but possibly makes the changes in predicted MHI-5 scores look more dramatic than they would with wider scaling. In Figure 1 below (Figure 1 in Response document), we showed only one graph and the y-axis extended to range from 40 to100. It still showed that the slopes of regression line for migrants ≤5 years since arrival was almost flat and all 95% CIs were overlapping. Now, we have revised the Results section in our manuscript by deleting the changes in the predicted MHI-5 scores to avoid possible confusion.

Figure 1. Relationships between skill discretion and mental health stratified by migrant status based on years since arrival

R2-3. The authors write (lines 329-330): Since the LR tests were not significant, the relationships between psychosocial job characteristics and mental health were not stratified by migrant status. Is the performed LR test the best way of evaluating effect measure modification? Do stratified analysis confirm the results of the LR test?

Response:

There are a number of ways to determine whether an interaction (effect modifier) is present, such as the LR test, global test (F statistic test), or comparing adjusted R squared value, BIC and AIC values between models with and without interaction term. We used the LR test in our study. We also conducted the other two tests; both of which indicated there was no significant effect-measure modification.

For example, the LR test result for whether years since arrival moderated the relationship between skill discretion and mental health was not significant (chi2(3)=1.32, p=0.72). The global test also reported a non-significant result (F(3,8953)=0.44, p=0.72). Finally, the adjusted R squared value was larger (0.0168 vs. 0.0166), and AIC and BIC were smaller (AIC: 75075.48 vs. 75080.16 and BIC: 75167.80 vs. 75193.78) for the model without interaction term. All three of the tests indicated that the model without interaction term was a better model.

The LR test provides evidence whether the stratification was justified or not, and stratification is not justified based on the results of the LR test in our study. We have revised the Results section according to the reviewer’s comments. We understood this revision included some discussion of the content, but we prefer to keep it in the Results section in the hope of clarifying why we did not stratify.

The corresponding revision to the text is (page 15-16, line 336-340):

Because none of the LR test results were statistically significant, there was no justification to stratify the psychosocial job characteristic―mental health relationships by migrant status. This suggests that the differences in the magnitude of the relationships between psychosocial job characteristic and mental health―which were also shown as the differences in the slopes of regression lines in Figs 2-4―between migrant subgroups and Australian-born workers were not statistically significantly different from zero.

R2-4. In this LR test, the interaction term consists of a psychosocial job characteristic variable, which is continuous, and a migrant status variable, which is categorical with 4 categories. Is the migrant status variable treated correctly in the interaction analysis? Often the variables in an interaction term are either continuous or dichotomous (as mentioned in ref. 26 in the response to the reviewers). How does the analysis treat the migrant status variable? As a categorical variable (with no intrinsic ordering to the categories)? As an ordinal variable (with clear ordering of the categories)? As a numerical variable (with equal intervals between the values of the variable)? Or as a continuous variable? Are all these alternatives possible in an LR analysis, e.g. categorical variables? Are all assumptions met for this analysis?

Response:

The migrant status in our study is a categorical variable without intrinsic ordering. The regression-based analysis for effect modification has no restriction on the nature of exposure variables and potential effect modifiers [3], and categorical variable can have more than two categories in testing effect modification [4]. 

R2-5. What do the partly overlapping confidence intervals indicate? If the constant is very similar but the slope very different, they would probably partly overlap, but does it mean that they are not significantly different?

Response:

The 95% confidence interval shows the upper and lower limits of the slope of regression lines of each subgroup based on migrant status. The large overlap indicates that these slopes are not statistically significantly different from each other. When there is effect modification, the lines may cross―which means the 95% CIs could overlap. The slopes would still be statistically significantly different from each other if the overlapping areas are not large. 

R2-6. If I understand correctly, Australian-born workers are the reference group that different migrant groups are compared to. However, this is not always clear.

Response:

Yes, Australian-born workers were always the reference group when compared with migrant workers in this study and thus, we did not state this specifically in each instance. However, we hypothesised that Non-ESC-born and newly arrived migrants would be the most vulnerable migrant subgroups, therefore, where we noticed clear differences between migrant subgroups, we would report that as well. We have now revised our manuscript to add the reference groups wherever possible. 

The corresponding revisions of the text are:

Introduction section (page 4, line 91-93)

However, results to the question of whether migrant workers are more sensitive to psychosocial job characteristics in terms of mental health compared to native-born workers are inconsistent.

Introduction section (page 5, line 116-117)

……, whether migrants are differentially affected by psychosocial job characteristic-associated impacts on mental health compared to Australian-born workers is unclear,……

Results section (page 14, line 311-315)

……, the regression lines for migrants who arrived ≤5 years previously appeared to differ from the lines for Australian-born workers and other migrant subgroups as well. The slopes of the regression lines for migrants who arrived ≤5 years previously were close to zero; however, their 95% CIs largely overlapped with that of Australian-born workers and other migrant subgroups.

Discussion section 

(page 16, line 346-349)

However, there was a suggestion that psychosocial job characteristic―mental health associations for migrants who had arrived in Australia ≤5 years previously differed from the associations for Australian-born workers and migrants who had been living in Australia for longer,……

(page 17, line 376-377)

Various explanations as to why migrants would not have a higher vulnerability to psychosocial job characteristics than native-born workers are plausible.

R2-7. TITLE: The title of the manuscript has been changed, however, I now find it difficult to read. The term "psychosocial job characteristic-associated impacts" describes the outcome, but is very long and complicated.

Response:

Thank you for this suggestion. We have now revised the Title.

The corresponding revision of the text is:

Psychosocial job characteristics and mental health: do associations differ by migrant status in an Australian working population sample?

R2-8. INTRODUCTION: Lines 52-57 (and elsewhere): The term "occupational health inequalities" is used to describe one or specific groups in this paragraph, without referring to which groups they are compared to. Inequalities/differences - compared to which groups?

Response:

In some situations, such as what we wrote in line 54-59, it was not difficult to understand the reference groups but difficult to mention all of them. Therefore, we did not list the reference groups one by one. Furthermore, with Australian-born workers always being the reference group in this study, we did not repeat it each time. We have incorporated the reviewer’s comment by adding the reference groups wherever possible in this revised manuscript.

The corresponding revisions of the text are:

Discussion section 

(page 19, line 412-413)

……, reducing job insecurity would seem to be the most appropriate target for reducing the risk of OHIs between migrant workers and Australian-born workers.

(page 19, line 415-416)

……, continuing language support may be helpful to reduce OHIs experienced by migrants compared to Australian-born workers.

R2-9. Line 75: Some of the psychosocial job characteristics are referred to as high or low, but this could also apply to job strain and job insecurity.

Response:

Thank you for this comment. We have revised this sentence in the Introduction section.

The corresponding revision to the text is (Page 4, line 75-78):

Adverse psychosocial job characteristics―including high job demands, low job control, high job strain, high job insecurity, effort-reward imbalance and lack of social support at work―have been shown to be associated with a wide range of adverse mental health outcomes……

R2-10. Lines 116-118: "We hypothesise that the psychosocial job characteristic―mental health relationships are stronger for migrants, especially … migrants recently arrived in Australia." Why do you hypothesise this? Isn't it contrary to studies you refer to?

Response:

Following the occupational health inequality (OHI) framework of Landsbergis et al. (1), disadvantaged population groups, such as migrants, are more likely to experience OHIs due to higher exposure and/or higher vulnerability. Moreover, previous studies have shown that disadvantaged population groups, such as workers in the lowest socioeconomic positions [5], were more vulnerable to psychosocial job characteristic-associated mental health impacts. 

However, findings of most previous studies showed that compared to native-born workers, migrant workers have similar or even weaker vulnerability to mental health impacts of exposure to psychosocial job characteristics. We were mindful of the shortcomings of the simplistic measures of migrant status used in most of the previously published literature. We considered that if years since arrival or language facility in the destination country were taken into account, such a measure of migrant status might show the hypothesised direction of relationship: 1) recent migrants could plausibly be less acculturated in the destination country, they would have had less time to develop social networks and supports to buffer the effects of stressors, and they might also be precarious in terms of housing and financial security; 2) migrants with low English skills would more vulnerable again because it might impede social integration and social connections, and non-Caucasian workers might also experience racism at work or in the general community. 

Working from general principles drawing from Landsbergis et al. (1) combined with our more detailed characterisation (and measures) of migrant status, we anticipated/hyothesised that we would see stronger exposure-outcome associations among migrant workers compared to native-born workers. We now have revised the Introduction section of our manuscript to explain our hypothesis establishment clearer.

The corresponding revisions to the text are:

(Page 4, line 79-83)

……Adverse psychosocial job characteristics……have been shown to be associated with a wide range of adverse mental health outcomes……However, the vulnerability to psychosocial job characteristic-associated mental health impacts may be different between working population groups―those who are disadvantaged have been reported to be more vulnerable. For example, compared to workers with higher socioeconomic status, those with lower socioeconomic status showed stronger associations between effort-reward imbalance and depression, as well as job strain and depression [5].

(Page 6, line 123-127)

Based on that migrant workers are usually characterised as a disadvantaged population group and thus may be more vulnerable to psychosocial job characteristic-associated mental health impacts, we hypothesise that the psychosocial job characteristic―mental health relationships are stronger for migrants, especially migrants from Non-English-speaking countries, and migrants recently arrived in Australia.

R2-11. MATERIAL AND METHODS: Line 191: "8983 observations answered…" Can an observation answer?

Response:

Thank you for this comment. We have revised this sentence.

The corresponding revision to the text is (Page 9, line 202-203):

……8983 observations had completed data on all the items of the three psychosocial job characteristic scales,……

R2-12. Lines 204-206: I suggest moving this sentence ("moreover, a small sample…") to the end of the paragraph, so that the last sentence ("This left 86 respondents…", lines 206-208) comes closer to the sentence "Among the 95 who refused to answer, 9 were missing all items…" (line 202), i.e. the 86 referred to in the current last sentence.

Response:

Thank you for this comment. We have revised this paragraph.

The corresponding revision to the text is (Page 10, line 214-218):

……Among the 95 who refused to answer, 9 were missing all items and 86 were missing one or two items for the three psychosocial job characteristic scales. The number who were missing one or two items for the psychosocial job characteristic scales (n=86) was much smaller with respect to the number who were missing all items (n=1184+286+9=1479) and thus, limiting possibilities for value substitution (e.g., mean substitution of single missing items).

R2-13. RESULTS: Lines 256-264: I suggest reversing the comparisons (if Australian-born workers are the reference). E.g., instead of "Australian-born workers included a significantly lower proportion of males than Main-ESC-born workers", I suggest: "Main-ESC-born workers included a significantly higher proportion of males than Australian-born workers".

Response:

Thank you for this comment. We have revised the comparisons.

The corresponding revision to the text is (Page 12, line 266-273):

The differences in gender, age and educational attainment between migrant workers and Australian-born workers were apparent (all p-values <0.001). For example, Main-ESC-born workers included a significantly higher proportion of males than Australian-born workers (57.3% vs. 50.1%). Migrants who arrived in Australia between 6 to 10 years previously included a significantly higher proportion of workers with a postgraduate degree compared to Australian-born workers (30.2% vs. 11.8%). In contrast, overseas-born workers included a significantly lower proportion of workers between 15 to 24 years of age than Australian-born workers (6.3% vs. 20.5%).

R2-14. In order to better understand and interpret the results, I would recommend adding a table showing the distribution of exposures and outcome (e.g. mean and range), according to migrant categories.

Response:

The research question in this study was to test whether migrant status modified the relationships between psychosocial job characteristics and mental health. The distributions of psychosocial job characteristics and mental health by migrant status were not related to the moderation test; thus, we believe this would be beyond the scope of our paper. Nevertheless, we have presented these results in a supplementary table (S1 Table). 

The corresponding revision to the text is (Page 13, line 278-279):

Distributions of psychosocial job characteristics and MHI-5 scores by migrant status were reported in supporting information S1 Table.

S1 Table. Distributions of psychosocial job characteristics and MHI-5 scores by migrant status

 Migrant status Minimum Maximum Median Mean (Standard error)

Skill discretion Australian-born 2 14 10 10.00 (2.76)

 Overseas-born 2 14 10 9.95 (2.73)

 Main-ESC-born† 2 14 10 10.14 (2.66)

 Non-ESC-born† 2 14 10 9.79 (2.78)

 Arrived ≤5 years 2 14 10 9.69 (2.95)

 Arrived 6-10 years 2 14 10 10.00 (2.94)

 Arrived ≥11 years 2 14 10 9.98 (2.67)

Decision authority Australian-born 3 21 12 12.44 (4.66)

 Overseas-born 3 21 13 13.06 (4.57)

 Main-ESC-born† 3 21 14 13.26 (4.72)

 Non-ESC-born† 3 21 13 12.89 (4.42)

 Arrived ≤5 years 3 21 12 11.91 (3.64)

 Arrived 6-10 years 3 21 13 12.50 (4.43)

 Arrived ≥11 years 3 21 14 13.27 (4.65)

Job insecurity Australian-born 3 21 8 8.47 (3.83)

 Overseas-born 3 21 9 8.91 (4.02)

 Main-ESC-born† 3 21 9 8.64 (3.89)

 Non-ESC-born† 3 21 9 9.14 (4.13)

 Arrived ≤5 years 3 21 10 10.00 (3.73)

 Arrived 6-10 years 3 21 9 9.28 (4.05)

 Arrived ≥11 years 3 21 9 8.74 (4.03)

MHI-5 score Australian-born 0 100 80 75.12 (16.00)

 Overseas-born 4 100 80 74.50 (16.24)

 Main-ESC-born† 4 100 80 75.12 (16.16)

 Non-ESC-born† 8 100 76 73.96 (16.30)

 Arrived ≤5 years 24 100 76 72.86 (16.56)

 Arrived 6-10 years 16 100 80 74.04 (16.04)

 Arrived ≥11 years 4 100 80 74.74 (16.23)

† Main-ESC-born: Born in Main-English-speaking country, Non-ESC-born: Born in Non-English-speaking country

R2-15. Lines 284-285: "since gender, age and educational attainment were not confounders for psychosocial job characteristic―mental health relationships". According to the model (S2 Fig), they were confounders, but they barely confounded the relationships, according to the results, i.e. they were only weak confounders.

Response:

In our study, these demographic factors only resulted in minimal changes to the coefficients (as shown in Table 2), therefore, it can be argued they were not confounders in this study, but they were included as covariates anyways for completeness. Because we have deleted the results regarding the changes in predicted MHI-5 scores with psychosocial job characteristics changing from the minimum to the maximum in this revised manuscript, the sentence the reviewer mentioned in this comment has been deleted as well.

R2-16. DISCUSSION: Lines 359-361: "… evidence that the most recently arrived migrants may not be sensitive to skill discretion and decision authority, suggesting associations for these two psychosocial job characteristics are stronger in Australian-born workers". You have not actually compared the two groups statistically, which I would recommend that you do.

Response:

In our manuscript, we wrote ‘We do, however, find suggestive, non-statistically significant evidence that the most recently arrived migrants may not be sensitive to skill discretion and decision authority, suggesting associations for these two psychosocial job characteristics are stronger in Australian-born workers (see Figs 2 and 3); however, these apparent differences may be due to the small sample size of the subgroup of migrants who arrived in Australia ≤5 years previously (n=154). Future study with a larger sample size would be required to resolve this question’ (page 17, line 369-375). We have presented this as suggestive and explained the possible explanation and the need for further study.

Our rationale for this discussion has been presented in the response to comment R2-1. Please see our response to R2-1.

R2-17. Lines 366-367: "Various explanations as to why migrants would not have a higher vulnerability to psychosocial job characteristics are plausible." You mention "healthy immigrant effect" and some other possible explanations. It is also worth mentioning that demographics of the most recently arrived migrants appear to be quite different, compared to the Australian-born workers: 55 % are 25-34 years of age (23 % of Australian-born), and 57 % are bachelor or postgraduates (29 % of Australian-born). Although you have adjusted for age and education level, these differences may also indicate other differences, which are not adjusted for. "Healthy immigrant effect" may partly cover this.

Response:

Thank you for this comment. The demographic factors could act as both confounders and effect modifiers for the psychosocial job characteristic―mental health relationships. In this study, we have controlled them as confounders, which indicated that their effects on the relationships could be seen as being “removed”. Thus, we prefer not to discuss their effects specifically. However, as the reviewer said, they could be effect modifiers as well, but we did not test this in the current study due to the small sample size of some subgroups. Accordingly, we have added this as a limitation in the Discussion section.

The corresponding revision to the text is (Page 20, line 437-441):

Some limitations should be considered in the interpretation of our findings……. Finally, our statistical power to assess effect-measure modification was limited by the relatively small sample sizes of subgroups, which not only resulted in the wide 95% CI of small subgroups but also prevented the three-way effect modification tests being conducted―in which the question of whether gender, age and educational attainment modified the moderation effect of migrant status on the relationships between psychosocial job characteristics and mental health would be tested.

R2-18. Lines 390-395: The sentence is very long and difficult to understand.

The statement: "Since migrants and Australian-born workers have similar sensitivity to psychosocial job characteristics in terms of mental health" is very categorical and does not quite harmonise with the results, in my opinion.

Response:

Thank you for this comment. We have revised this sentence now.

The corresponding revision to the text is (Page 18, line 401-402):

Since we did not find statistically significant evidence that the impacts of psychosocial job characteristics on mental health were different between migrant and Australian-born workers……

R2-19. Lines 417-420: In what direction would common method bias probably affect the results? Usually this type of bias leads to stronger associations, which could be added.

Response:

Thank you for this comment. We have revised this limitation.

The corresponding revision to the text is (Page 20, line 431-432):

……thus common method bias could potentially inflate the relationship between psychosocial job characteristics and mental health.

R2-20. Lines 420-424: This could also make it difficult to observe differences between migrants from Non-English-speaking countries and English-speaking countries.

Response:

Thank you for this comment. We have revised this limitation now.

The corresponding revision to the text is (Page 20, line 434-436):

This exposure misclassification would make it harder to observe the differences in psychosocial job characteristic―mental health associations between migrants from Non-English-speaking countries and both native-born workers and migrants from Main-English-speaking countries.

R2-21. REFERENCES: Check Reference 49 (Laura).

Response:

Thank you for this comment. We have deleted this reference.

References

1. Landsbergis PA, Grzywacz JG, LaMontagne AD. Work organization, job insecurity, and occupational health disparities. Am J Ind Med. 2014;57(5):495-515. doi: 10.1002/ajim.22126. PubMed PMID: 23074099.

2. Karasek RA. Job demands, job decision latitude, and mental strain: Implications for job redesign. Adm Sci Q. 1979;24(2):285-308. doi: Doi 10.2307/2392498. PubMed PMID: WOS:A1979GX41100007.

3. Hayes AF. Introduction to mediation, moderation, and conditional process analysis: A regression-based approach. Second ed. New York: Guilford publications; 2018. 223-66 p.

4. UCLA. How can i understand a categorical by continuous interaction? (Stata 12) | Stata FAQ 2020 [cited 2020 28 July]. Available from: https://stats.idre.ucla.edu/stata/faq/how-can-i-understand-a-categorical-by-continuous-interaction-stata-12/.

5. Wege N, Dragano N, Erbel R, Jockel KH, Moebus S, Stang A, et al. When does work stress hurt? Testing the interaction with socioeconomic position in the Heinz Nixdorf Recall Study. J Epidemiol Community Health. 2008;62(4):338-41. doi: 10.1136/jech.2007.062315. PubMed PMID: 18339827.

---

## [Decision Letter · Decision Letter 2]

7 Oct 2020

PONE-D-19-35389R2

Psychosocial job characteristics and mental health: do associations differ by migrant status in an Australian working population sample?

PLOS ONE

Dear Dr. LIU,

Thank you for submitting your manuscript to PLOS ONE. After careful consideration, we feel that it has merit but does not fully meet PLOS ONE’s publication criteria as it currently stands. Therefore, we invite you to submit a revised version of the manuscript that addresses the points summarised by me. The revised manuscript will be reviewed by me rather than receiving another round of reviews. To be successful, the revision must include the following:

A more concrete answer to Reviewer 2's question in comment R2-3 "Is the performed LR tests the best way to assess effect measure modification?" comparing the various approaches availableThe implementation of the recommendations by Knol and VanderWeele (2012) were relevant; if at all, it should be further outlined why recommendations were not followedA more careful application of the principle "Absence of evidence is not the same as evidence of absence" (as suggested by Reviewer 2)A revised first sentence of the abstractA revision of the following anthropomorphism in line 202: "8983 observations had completed data"Table S1 in the manuscript

We look forward to receiving your revised manuscript.

Kind regards,

Dana Unger

Academic Editor

PLOS ONE

Reviewers' comments:

Reviewer's Responses to Questions

**Comments to the Author**

1. If the authors have adequately addressed your comments raised in a previous round of review and you feel that this manuscript is now acceptable for publication, you may indicate that here to bypass the “Comments to the Author” section, enter your conflict of interest statement in the “Confidential to Editor” section, and submit your "Accept" recommendation.

Reviewer #2: (No Response)

2. Is the manuscript technically sound, and do the data support the conclusions?

Reviewer #2: Partly

3. Has the statistical analysis been performed appropriately and rigorously? 

Reviewer #2: I Don't Know

4. Have the authors made all data underlying the findings in their manuscript fully available?

Reviewer #2: No

5. Is the manuscript presented in an intelligible fashion and written in standard English?

Reviewer #2: Yes

6. Review Comments to the Author

Reviewer #2: I have reviewed this manuscript twice previously.

Thank you for the revised manuscript and responses to my comments.

The authors have responded adequately to some of my comments. However, I still have comments to the analyses and how the results are interpreted, which is still my major concern.

In my view, the authors make some very common mistakes, which have been discussed by leading epidemiologists, e.g. in the following publications:

1. Schmidt M, Rothman KJ. Mistaken inference caused by reliance on and misinterpretation of a significance test. Int J Cardiol. 2014;177:1089–90.

2. Knol MJ, Pestman WR, Grobbee DE. The (mis)use of overlap of confidence intervals to assess effect modification. Eur J Epidemiol. 2011 Apr;26(4):253-4. doi: 10.1007/s10654-011-9563-8.

And they do not follow the recommendations, in e.g. this publication:

3. Knol MJ, VanderWeele TJ. Recommendations for presenting analyses of effect modification and interaction. Int J Epidemiol. 2012 Apr;41(2):514-20. doi: 10.1093/ije/dyr218.

I will comment some of the answers to my previous comments:

R2-1

“…the statistical evidence suggested no effect modification”.

Absence of evidence is not the same as evidence of absence. It is not statistically significant, but that does not mean there is no effect modification. See reference 1 above.

“…testing effect modification is different from stratification”.

That is true. However, leading epidemiologists recommend presenting analyses of effect modification as stratified analyses. See reference 3 above.

Table 1 in the answer clearly shows that differences in coefficients between migrants and Australian-born are much larger than for the other groups, although not statistically significant.

R2-2

The changes in predicted MHI-5 scores with psychosocial job characteristics changing from the minimum to the maximum is the same (equal to) the slopes of the regression lines in unadjusted analyses.

“all 95% CIs were overlapping”.

This does not mean there is no effect modification. See reference 2 above.

R2-3

As mentioned, leading epidemiologists recommend presenting analyses of effect modification as stratified analyses. See reference 3 above.

R2-8

The response to my comment was adequate but still, the first sentence of the abstract is written as follow:

“Migrant workers are more likely to experience occupational health inequalities than native-born workers”.

Do native-born workers experience occupational health inequalities? In comparison to whom?

R2-14

S1 Table is important, as it shows why there is effect modification for newly arrived migrants; their mean scores for psychosocial factors and mental health differ from the other migrant groups and Australian-born people.

7. PLOS authors have the option to publish the peer review history of their article (what does this mean?). If published, this will include your full peer review and any attached files.

Reviewer #2: No

---

## [Author Response · Author response to Decision Letter 2]

1 Nov 2020

Academic Editor

We invite you to submit a revised version of the manuscript that addresses the points summarised by me. The revised manuscript will be reviewed by me rather than receiving another round of reviews. To be successful, the revision must include the following:

AE-1: A more concrete answer to Reviewer 2's question in comment R2-3 "Is the performed LR tests the best way to assess effect measure modification?" comparing the various approaches available

Response:

We have conducted our effect measure modification analyses using standard legitimate methods. Whether it is the “best way” is subject to debate—the key question is whether our approach is legitimate. Below, we outline some of the alternative methods, each of which has also been applied in our analyses, yielding—as expected—qualitatively identical results indicating a lack of statistical evidence for effect measure modification.

The fundamental approach is to fit a product term (exposure × effect modifier) and compare the model fit with versus without the product term in the model. Then “various approaches” available could refer to the choice of test statistics to compare the model fit. In Stata, the test statistic used for the likelihood ratio test (lrtest) is a Chi squared. Another approach applies a global F-test. Both tests provide similar p-value results, and a similar interpretation of the p-value can be made. Model fit can also be compared using Aikake’s Information Criteria (AIC) and Bayesian Information Criteria (BIC). These tests provide the same results again for our research question—no statistical evidence to indicate effect measure modification (these results have been provided in previous responses to review). 

In many journals, a null result for comparative model fit testing is adequate in itself, and this is an adequate basis for ruling out effect measure modification in the case at hand and not presenting stratified results. Realising that we had two relatively small groups in our effect measure modification analyses (migrants resident ≤5 years [n=154] and 6-10 years [n=199]), we decided to present graphical results because the small sample sizes would limit statistical power and graphical results might suggest differences between groups (a clear demonstration of our awareness that ‘absence of [statistical] evidence is not evidence of absence’). This would be an important finding to inform future studies of the need for larger sample sizes. This is where the trouble with this paper began, because the slope for some results among migrants who had been resident for ≤5 years looked like they might be different from others, despite the null statistical results. 

The ensuing debate with Reviewer 2 led us to Knol and VanderWeele (2012) and the ongoing debates in the literature on how best to present effect modification analysis results. This has been informative, but Knol and VanderWeele’s recommendations for presenting results on the relative magnitude and direction of differences in exposure-outcome relationships across effect modifier groups apply only to studies using binary or categorical outcome and exposure measures yielding Odds Ratios or Relative Risk measures (e.g., case-control studies). To achieve the same end of presenting the relative magnitude and direction of effect measure modification results in our paper, we have now added an analysis of differences in beta coefficients or slopes by each stratum of the relevant effect modifier, following on from a recommendation by Knol, Pestman and Grobbee (2011), cited by Reviewer 2, that “other methods to assess effect modification could be used, such as calculating a 95% confidence interval around the difference in effect estimates.” This was done in Stata using Pairwise comparisons of average marginal effects. We have revised the manuscript now to present these results in Tables 3-5 (below). In summary, these analyses show that none of the differences in effect estimates between strata of any of the three effect measure modifiers tested were statistically significant: all of the 95% CIs comparing slope differences in Tables 3-5 include zero. The figures which were previously presented in the main manuscript have now been shifted to Supporting Information and resized to increase readability (S3-S5 Figs, also included at the end of this document).

In summary, our effect measure modification analysis now includes 1) comparative model fit testing, 2) analysis of differences in effect estimates by strata of effect modifiers, and 3) descriptive presentation of graphical results. The first two analyses include formal statistical testing, and the third is for information (hence has been placed with Supporting Information). In this way, we provide both a global assessment of effect measure modification (analysis 1) as well as detail on the magnitude and direction of results within strata of effect modifiers (analyses 2 and 3).

The corresponding revisions to the text are:

Methods

(Page 10-11, lines 228-240)

Third, effect measure modification of the job characteristic―mental health relationship by migrant status was tested by fitting product terms between psychosocial job characteristics and migrant status and comparing model fit between the model with versus without the product term using the Likelihood Ratio (LR) test. Considering that LR tests only provide measures of statistical significance without information on the relative magnitude and direction of differences in exposure-outcome relationships across effect modifier groups, comparative model fit testing was complemented for descriptive/explanatory purposes by graphing the relationships between psychosocial job characteristics and mental health by migrant status with 95% confidence intervals (CIs). In addition, the differences in the slope estimates of regression lines between effect modifier groups were assessed by average marginal effects comparisons [45]. LR test results or differences in slope estimates at p<0.05 was considered to constitute statistical evidence of effect measure modification and would justify the stratified presentation of job characteristic―mental health relationships as main findings.

Results

(Page 14, lines 299-300)

Likelihood ratio test results were uniformly null, providing no statistical evidence for any of our three measures of migrant status as effect modifiers of the job characteristic―mental health relationships……

(Page 14, lines 306-312)

When graphing effect measure modification results, it appeared that slopes for migrants who had been resident for ≤5 years might differ from other groups for both decision authority and skill discretion analyses (see S3 and S4 Figs; slope estimates are reported in Tables 3 and 4). We would also note the recently arrived migrant group (≤5 years resident) was the smallest of all migrant subgroups analysed (n=154, Table 1), and thus yielded the least precise beta/slope estimates. Consistent with the LR test results, Tables 3 and 4 show that all of the 95% CIs comparing slope differences for skill discretion and decision authority analyses included zero.

Table 3. Relationships between skill discretion and mental health stratified by three measures of migrant status: the slope estimate within each stratum of migrant status and the differences in the slope estimates between groups, controlled for gender, age and educational attainment.

 Predicted mental health (MHI-5 score)

 Migrant status measure one Migrant status measure two Migrant status measure three

Slope estimates within the stratum of migrant status Migrant status Coef. (95% CI) Migrant status Coef. (95% CI) Migrant status Coef. (95% CI)

 Australian-born 0.38 (0.24, 0.51)*** Australian-born 0.38 (0.24, 0.51)*** Australian-born 0.38 (0.24, 0.51)***

 Overseas-born 0.48 (0.20, 0.75)*** Main-ESC-born† 0.37 (-0.04, 0.78) Arrived ≤5 years 0.10 (-0.76, 0.95)

 Non-ESC-born† 0.55 (0.19, 0.92)** Arrived 6-10 years 0.59 (-0.16, 1.34)

 Arrived ≥11 years 0.51 (0.20, 0.83)***

Differences in slope estimates Overseas-born vs. Australian-born 0.10 (-0.20, 0.41) Main-ESC-born† vs. Australian-born -0.01 (-0.44, 0.43) Arrived ≤5 years vs. Australian-born -0.28 (-1.15, 0.58)

 Non-ESC-born† vs. Australian-born 0.18 (-0.21, 0.57) Arrived 6-10 years vs. Australian-born 0.21 (-0.55, 0.98)

 Non-ESC-born† vs. Main-ESC-born† 0.18 (-0.37, 0.74) Arrived ≥11 years vs. Australian-born 0.13 (-0.21, 0.48)

 Arrived 6-10 years vs. Arrived ≤5 years 0.49 (-0.64, 1.63)

 Arrived ≥11 years vs. Arrived ≤5 years 0.42 (-0.49, 1.32)

 Arrived ≥11 years vs. Arrived 6-10 years -0.08 (-0.89, 0.74)

Migrant status measure one: country of birth (COB), measure two: COB and English/Non-English dominant language of COB, measure three: COB and years since arrival; † Main-ESC-born: Born in a Main-English-speaking country; Non-ESC-born: Born in a Non-English-speaking country; Coef. was the beta coefficient; *** p<0.001, **p<0.01, *p<0.05.

Table 4. Relationships between decision authority and mental health stratified by three measures of migrant status: the slope estimate within each stratum of migrant status and the differences in the slope estimates between groups, controlled for gender, age and educational attainment.

 Predicted mental health (MHI-5 score)

 Migrant status measure one Migrant status measure two Migrant status measure three

Slope estimates within the stratum of migrant status Migrant status Coef. (95% CI) Migrant status Coef. (95% CI) Migrant status Coef. (95% CI)

 Australian-born 0.36 (0.28; 0.44)*** Australian-born 0.36 (0.28; 0.44)*** Australian-born 0.36 (0.28; 0.44)***

 Overseas-born 0.37 (0.21; 0.54)*** Main-ESC-born† 0.27 (0.03; 0.50)* Arrived ≤5 years -0.04 (-0.73; 0.65)

 Non-ESC-born† 0.47 (0.24; 0.70)*** Arrived 6-10 years 0.58 (0.08; 1.08)*

 Arrived ≥11 years 0.37 (0.19; 0.55)***

Differences in slope estimates Overseas-born vs. Australian-born 0.01 (-0.17; 0.19) Main-ESC-born† vs. Australian-born -0.09 (-0.34; 0.15) Arrived ≤5 years vs. Australian-born -0.40 (-1.10; 0.29)

 Non-ESC-born† vs. Australian-born 0.11 (-0.13; 0.36) Arrived 6-10 years vs. Australian-born 0.22 (-0.28; 0.73)

 Non-ESC-born† vs. Main-ESC-born† 0.21 (-0.12; 0.53) Arrived ≥11 years vs. Australian-born 0.01 (-0.18; 0.21)

 Arrived 6-10 years vs. Arrived ≤5 years 0.62 (-0.23; 1.48)

 Arrived ≥11 years vs. Arrived ≤5 years 0.42 (-0.30; 1.13)

 Arrived ≥11 years vs. Arrived 6-10 years -0.21 (-0.74; 0.32)

Migrant status measure one: country of birth (COB), measure two: COB and English/Non-English dominant language of COB, measure three: COB and years since arrival; † Main-ESC-born: Born in a Main-English-speaking country; Non-ESC-born: Born in a Non-English-speaking country; Coef. was the beta coefficient; *** p<0.001, **p<0.01, *p<0.05.

(Page 15, lines 328-332)

In the job insecurity―mental health relationships, the regression lines for all groups, including Australian-born workers and all migrant subgroups based on COB, dominant language of COB and years since arrival in Australia, were almost overlapping (see S5 Fig; slope estimates are reported in Table 5). Consistently, Table 5 shows none of the differences in slope estimates between groups was significantly different from zero. 

Table 5. Relationships between job insecurity and mental health stratified by three measures of migrant status: the slope estimate within each stratum of migrant status and the differences in the slope estimates between groups, controlled for gender, age and educational attainment.

 Predicted mental health (MHI-5 score)

 Migrant status measure one Migrant status measure two Migrant status measure three

Slope estimates within the stratum of migrant status Migrant status Coef. (95% CI) Migrant status Coef. (95% CI) Migrant status Coef. (95% CI)

 Australian-born -1.08 (-1.18; -1.00)*** Australian-born -1.08 (-1.18; -1.00)*** Australian-born -1.08 (-1.18; -1.00)***

 Overseas-born -1.03 (-1.21; -0.85)*** Main-ESC-born† -0.99 (-1.26; -0.71)*** Arrived ≤5 years -0.79 (-1.44; -0.14)*

 Non-ESC-born† -1.07 (-1.31; -0.83)*** Arrived 6-10 years -1.27 (-1.80; -0.74)***

 Arrived ≥11 years -1.03 (-1.24; -0.83)***

Differences in slope estimates Overseas-born vs. Australian-born 0.05 (-0.15; 0.25) Main-ESC-born† vs. Australian-born 0.10 (-0.19; 0.39) Arrived ≤5 years vs. Australian-born 0.29 (-0.37; 0.95)

 Non-ESC-born† vs. Australian-born 0.01 (-0.25; 0.27) Arrived 6-10 years vs. Australian-born -0.18 (-0.72; 0.35)

 Non-ESC-born† vs. Main-ESC-born† -0.09 (-0.45; 0.28) Arrived ≥11 years vs. Australian-born 0.05 (-0.17; 0.27)

 Arrived 6-10 years vs. Arrived ≤5 years -0.48 (-1.32; 0.36)

 Arrived ≥11 years vs. Arrived ≤5 years -0.24 (-0.93; 0.44)

 Arrived ≥11 years vs. Arrived 6-10 years 0.23 (-0.33; 0.80)

Migrant status measure one: country of birth (COB), measure two: COB and English/Non-English dominant language of COB, measure three: COB and years since arrival; † Main-ESC-born: Born in a Main-English-speaking country; Non-ESC-born: Born in a Non-English-speaking country; Coef. was the beta coefficient; *** p<0.001, **p<0.01, *p<0.05.

(Page 16, lines 341-345)

In summary, there was no statistical evidence supporting any of our three measures of migrant status as effect modifiers of the relationships between psychosocial job characteristics and mental health, despite the suggestion of some differences by visual inspection of graphical results. The main finding of our effect measure modification analysis is that the relationship between the three job characteristics and mental health does not differ by any of the three measures of migrant status used.

Discussion

(Page 18, lines 380-383)

We did, however, find suggestive evidence that skill discretion and decision authority may not be associated with the mental health of most recently arrived migrants, suggesting associations for these two psychosocial job characteristics are stronger in Australian-born workers (see Tables 3 and 4 and S3 and S4 Figs);……

AE-2: The implementation of the recommendations by Knol and VanderWeele (2012) were relevant; if at all, it should be further outlined why recommendations were not followed.

Response:

This has now been answered above in our response to AE-1. To summarise:

The specific recommendations for presenting relative excess risk due to interaction (RERI) in effect modification results by Knol and VanderWeele (2012) apply only to situations where the outcome, exposure, and effect modifier are binary or categorical variables (e.g., case-control studies). In our paper, we are relating continuous measures of outcome and exposure, and assessing that relationship for effect measure modification by binary and categorical measures of migrant status.

To provide detail on the relative magnitude and direction of effect measure modification results in our paper (the purpose of Knol & VanderWeele’s recommendations to provide RERI or Synergy Index results for binary or categorical measures), we have assessed differences in effect estimates with 95% CIs across strata of effect modifiers (Knol, Pestman & Grobbee 2011), using pairwise comparisons of average marginal effects, as detailed in our preceding response.

AE-3: A more careful application of the principle "Absence of evidence is not the same as evidence of absence" (as suggested by Reviewer 2).

Response:

We believe the reviewer and we were sharing the same recognition that there may be effect measure modification that our study was underpowered to detect. A non-critical acceptance of our comparative model fit testing results would have precluded presentation of graphical effect modification results. This clearly demonstrates that we believe that “absence of evidence is not the same as evidence of absence.” We hope this is now clear.

AE-4: A revised first sentence of the abstract.

Response:

We have revised this sentence.

The corresponding revision to the text is (page 2, line 32-33):

Migrant workers may experience higher burdens of occupational injury and illness compared to native-born workers,……

AE-5: A revision of the following anthropomorphism in line 202: “8983 observations had completed data”.

Response:

Thank you for this comment. We have revised this sentence.

The corresponding revision to the text is (page 9, line 201-202):

Among these 10569, 8983 observations had complete items for the three psychosocial job characteristic scales,……

AE-6: Table S1 in the manuscript (Reviewer’s comment: S1 Table is important, as it shows why there is effect modification for newly arrived migrants; their mean scores for psychosocial factors and mental health differ from the other migrant groups and Australian-born people).

Response:

S1 Table provides univariate descriptive information (distributions, summary measures) on psychosocial factors and MHI-5 scores for Australian-born workers and each migrant subgroup, but nothing on exposure-outcome associations. Nothing about the associations between job characteristics and mental health, nor difference in those associations by migrant status can be concluded from this table.

Reference

Knol, MJ, Pestman, WR & Grobbee, DE 2011, 'The (mis) use of overlap of confidence intervals to assess effect modification', European Journal of Epidemiology, vol. 26, no. 4, pp. 253-4.

Knol, MJ & VanderWeele, TJ 2012, 'Recommendations for presenting analyses of effect modification and interaction', International Journal of Epidemiology, vol. 41, no. 2, pp. 514-20.

---

## [Editor Report · Decision Letter 3]

12 Nov 2020

Psychosocial job characteristics and mental health: do associations differ by migrant status in an Australian working population sample?

PONE-D-19-35389R3

Dear Dr. LIU,

We’re pleased to inform you that your manuscript has been judged scientifically suitable for publication and will be formally accepted for publication once it meets all outstanding technical requirements. The study is very relevant and should advance the field. Thank you for your patience and for engaging in the review process with us in the way you did. 

Kind regards,

Dana Unger

Academic Editor

PLOS ONE

---

## [Editor Report · Acceptance letter]

16 Nov 2020

PONE-D-19-35389R3 

Psychosocial job characteristics and mental health: do associations differ by migrant status in an Australian working population sample? 

Dear Dr. Liu:

I'm pleased to inform you that your manuscript has been deemed suitable for publication in PLOS ONE. Congratulations! Your manuscript is now with our production department. 

Kind regards, 

on behalf of

Dr. Dana Unger 

Academic Editor

PLOS ONE